# Using cellular fitness to map the structure and function of a major facilitator superfamily effluxer

Anisha M Perez[1],[†], Marcella M Gomez[2],[†], Prashant Kalvapalle[3], Erin O'Brien-Gilbert[1], Matthew R Bennett[1],[4] & Yousif Shamoo[1],[*] (iD)

## Abstract

The major facilitator superfamily (MFS) effluxers are prominent mediators of antimicrobial resistance. The biochemical characterization of MFS proteins is hindered by their complex membrane environment that makes *in vitro* biochemical analysis challenging. Since the physicochemical properties of proteins drive the fitness of an organism, we posed the question of whether we could reverse that relationship and derive meaningful biochemical parameters for a single protein simply from fitness changes it confers under varying strengths of selection. Here, we present a physiological model that uses cellular fitness as a proxy to predict the biochemical properties of the MFS tetracycline efflux pump, TetB, and a family of single amino acid variants. We determined two lumped biochemical parameters roughly describing $K_m$ and $V_{max}$ for TetB and variants. Including *in vivo* protein levels into our model allowed for more specified prediction of pump parameters relating to substrate binding affinity and pumping efficiency for TetB and variants. We further demonstrated the general utility of our model by solely using fitness to assay a library of *tet(B)* variants and estimate their biochemical properties.

**Keywords** antibiotic resistance; efflux pump; major facilitator superfamily; structure function; tetracycline

**Subject Categories** Microbiology, Virology & Host Pathogen Interaction; Pharmacology & Drug Discovery; Quantitative Biology & Dynamical Systems

**Mol Syst Biol. (2017) 13: 964**

## Introduction

Determining the link between protein function and fitness is qualitatively simple to understand as the fitness of an organism is dependent on proteins encoded by the genome. However, even the modestly sized genome of *Escherichia coli* consists of thousands of genes making deconvolution of protein changes and their effects on fitness a challenging problem (Blattner *et al*, 1997). Under strong selection conditions, the complexity of this problem can be decreased by tightly linking the performance of a single gene product to fitness, allowing for quantitative understanding of protein function and organismal fitness (Dean *et al*, 1986; Couñago *et al*, 2006; Walkiewicz *et al*, 2012). Typically, well-behaved proteins whose function is strongly correlated to organismal fitness are used to study this link as their biochemical properties can be readily measured *in vitro*. Such model proteins include TEM-1 β-lactamase, dihydrofolate reductase, and adenylate kinase (Wang *et al*, 2002; Couñago *et al*, 2006; Weinreich *et al*, 2006; Peña *et al*, 2010; Jacquier *et al*, 2013; Rodrigues *et al*, 2016). These advances have provided a greater understanding into how the physicochemical properties of a protein relate to fitness and allow investigators to explore the role of fitness landscapes in adaptive evolution (Weinreich *et al*, 2006; Dean & Thornton, 2007; Walkiewicz *et al*, 2012; Harms & Thornton, 2013; Meini *et al*, 2015; Palmer *et al*, 2015).

Implicit in the idea that the *in vitro* properties of a protein can be used to predict fitness (physicochemical properties → phenotype) is the notion that fitness over a range of selection conditions could also be used to estimate the biochemical properties of a protein whose *in vitro* characterization is difficult or unknown (phenotype → physicochemical properties). For this study, we define fitness as the growth rate of the cells in exponential phase and the fitness function to be the growth rate of cells as a function of selection strength. We previously reported the development of a mathematical model that predicts the fitness of an organism at selective conditions from the *in vitro* physicochemical properties of a tetracycline resistance protein, TetX2, and a family of variants (Walkiewicz *et al*, 2012). During exposure to minocycline (MCN), the performance of TetX2 is tightly linked to the fitness of those cells. Using *in vitro* physicochemical properties and *in vivo* protein expression measurements, the model was able to predict the shape of the fitness function curves. Here, we utilize the fundamental reversibility of the TetX2 model and present a new *in vivo* model capable of determining relative biochemical properties of an efflux

1   Department of Biosciences, Rice University, Houston, TX, USA
2   Department of Applied Mathematics & Statistics, University of California, Santa Cruz, CA, USA
3   Systems, Synthetic, and Physical Biology Graduate Program, Rice University, Houston, TX, USA
4   Department of Bioengineering, Rice University, Houston, TX, USA
    *Corresponding author. Tel: +1 713 348 5493; E-mail: shamoo@rice.edu
    †These authors contributed equally to this work

transporter from fitness without the need for *a priori in vitro* protein analysis.

Integral membrane proteins are difficult to characterize using classical *in vitro* biochemical techniques as they are challenging to express, purify, and often require recapitulation of the membrane system for the reconstitution of activity. Analysis of this protein class, thus, provides an excellent opportunity for evaluation of our ability to derive useful biochemical properties from a fitness function. As a model system, we used the tetracycline resistance efflux pump *tet(B)* to examine whether growth rates of bacteria carrying a variant allele over a range of selection strength and an appropriate model can be used to determine physicochemical parameters such as substrate binding rate ($k_1$), and pumping efficiency ($r$). TetB is a 12-transmembrane alpha helical protein, which traverses the inner membrane of Gram-negative bacteria and uses the proton motive force to efflux members of the tetracycline family of antibiotics including, tetracycline (TET), doxycycline (DOX), and MCN in exchange for $H^+$. TetB is a member of the major facilitator superfamily (MFS) of transporters, the largest class of secondary transporters responsible for a wide variety of physiological processes such as: cell homeostasis, nutrient import, and toxin export (Pao *et al*, 1998; Yan, 2013). In addition to TetB, there are many other MFS transporters, which contribute to single and multidrug resistance in a wide variety of pathogens and their thorough characterization is of great interest (Fluman & Bibi, 2009; Alegre *et al*, 2016). Despite the low percentage of protein sequence similarity among MFS transporters, their overall structure is largely conserved with a general understanding of how specific regions of the transporter contribute to substrate binding and/or pumping efficiency (Yan, 2015; Quistgaard *et al*, 2016).

Using only fitness functions and *in vivo* protein levels from strains expressing *tet(B)* and *tet(B)* variants, we were able to construct a simple but robust model capable of predicting physicochemical properties describing substrate binding rate and pumping efficiency of TetB and variants. For these variants, our data are in good agreement with the current knowledge of how MFS transporter structure determines function. Our success in determining quantitative physicochemical properties from fitness at varying selective conditions suggests that with the appropriate mathematical model, libraries of protein variants can be rapidly screened without the need for individual protein purification or that small molecule libraries of potential inhibitors of those MFS family members associated with antibiotic resistance could be screened rapidly and useful biochemical properties ascribed without prior *in vitro* characterization.

# Results

### Baseline response of a cellular system without *tet(B)* is used to determine global parameters

While the qualitative concept of cellular phenotype as a product of protein(s) function is clear, the ability to deconstruct and accurately map the phenotype to physicochemical properties of a specific protein within the larger cellular milieu is an interesting challenge (Tan *et al*, 2012; Walkiewicz *et al*, 2012). We describe a general

mathematical model that allows us to relate changes in fitness function to changes in biophysical behaviors and could be applied to any effluxer that contributes strongly to fitness over a set of environmental conditions. To establish the global parameters required for the overall system, we measured the growth rate of the host cell, *E. coli* BW25113 without the effluxer *tet(B)* under a range of environmental selection conditions which for TetB are the antibiotics TET, DOX, and MCN. These fitness functions comprise the baseline response of the cellular system, which include any potential effects from other pumps or resistance mechanisms, and allow us to isolate the effects of *tet(B)* on fitness when it is added to the system. Without expression of *tet(B)*, we assume that diffusion is the major factor influencing the steady-state concentration of antibiotic in the cytoplasm. Previous work has shown that for TET the concentration of drug in the cytoplasm, $[S]_C$, is about four times that in the media, $[S]_M$, (Thanassi *et al*, 1995). Both DOX and MCN are more lipophilic than TET, which may lead to differences in their diffusion patterns; however, for the model presented here we take this same diffusion relationship for DOX and MCN (Fig 1A). The fitness function curves were fit to the following Hill-function relation that describes growth rate inhibition by cytoplasmic drug concentration

$$GR = 1 - \frac{[S]_C^B}{A^B + [S]_C^B}, \tag{1}$$

where $[S]_C$ is the concentration of substrate in the cytoplasm and $A$ and $B$ are global parameters that describe our specific cellular system (Fig 1, Table 1 and Code EV2). The tetracycline antibiotics inhibit growth by binding to the 30S ribosomal subunit. This interaction is represented through $A$, which corresponds to the cytoplasmic drug concentration, $[S]_C$, at which the normalized growth rate is half maximal. Additionally, global parameter $A$ can roughly be interpreted as the apparent $K_d$ of the antibiotic binding to the ribosome although it does not take into account other cellular sinks of the antibiotic (Walkiewicz *et al*, 2012). Global parameter $B$ is the Hill coefficient that describes the shape of the dose–response curve in response to antibiotic. It should be noted that our host strain *E. coli* BW25113, in contrast to many *E. coli* strains, has a higher intrinsic resistance to DOX and MCN than TET (Fig 1B). This intrinsic resistance, however, is captured by our model and reflected in parameter $A$ where this value is largest for MCN and smallest of TET (Table 1).

### A physiological model describing TetB efflux pump kinetics can be used to determine physicochemical parameters from fitness

Determining physicochemical properties from fitness at selective conditions requires the use of an appropriate physiological model. To do this, we mathematically modeled how a MFS efflux pump counteracts substrate diffusion *in vivo*. As an antiporter, TetB reduces the concentration of antibiotic in the cytoplasm by utilizing the proton motive force to pump one proton into the cytoplasm and pump out one tetracycline-$Mg^{1+}$ complex into the periplasm. Analysis of the BW25113 pump-free system allowed us to directly predict the concentration of antibiotic in the cytoplasm by the measured growth rate (equation 1). We can determine the substrate binding rate and pumping efficiency necessary to achieve the predicted

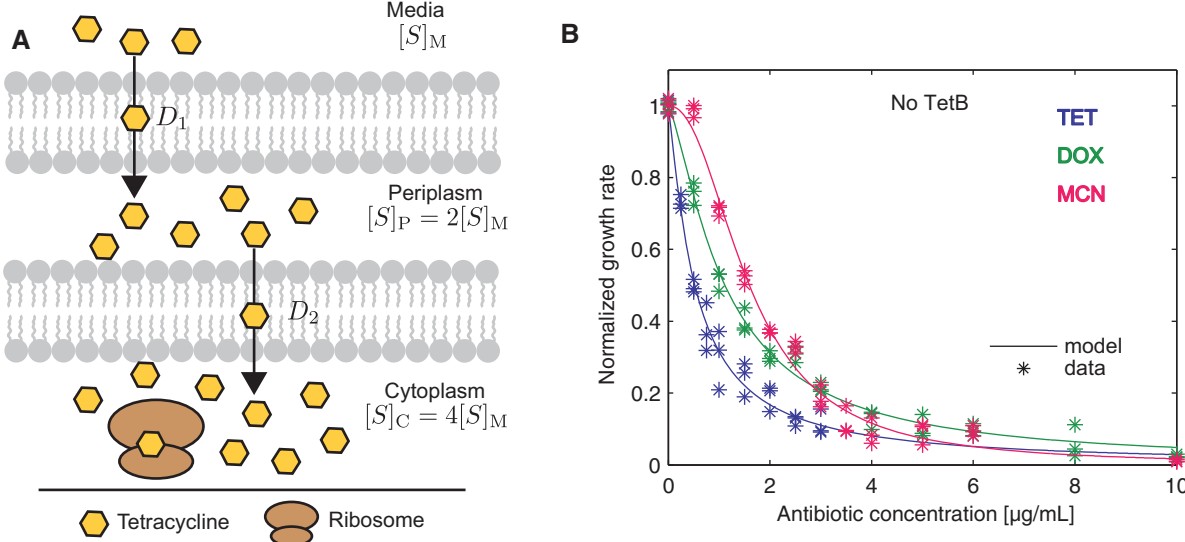

**Figure 1. Model incorporating substrate diffusion can be used to determine global parameters that describe the baseline response of the cellular system at a range of environmental selection conditions.**

A   Diffusion model including inhibition of growth rate by tetracycline antibiotics. The diffusion pattern presented is specific for TET in *Escherichia coli*. DOX and MCN are second-generation tetracycline antibiotics, which were created to have greater potency. For this model, however, we assume that the TET diffusion model is comparable to DOX and MCN.

B   Normalized fitness functions of *E. coli* BW25113 in the absence of *tet(B)* and their corresponding model fits in the presence of varying concentrations of antibiotics TET, DOX, and MCN in the media ($[S]_M$). Normalized quadruplicate data are shown for each antibiotic.

Source data are available online for this figure.

---

**Table 1. Global parameters *A* and *B*, which describe the baseline response of the cellular system, were computed from fitness functions of *Escherichia coli* BW25113 in TET, DOX, and MCN using equation (1).**

|       | Max  | Min  | Average | LS fit | SD   |
|-------|------|------|---------|--------|------|
| TET   |      |      |         |        |      |
| A     | 2.68 | 1.9  | 2.29    | 2.16   | 0.39 |
| B     | 1.02 | 1.37 | 1.20    | 1.15   | 0.18 |
| DOX   |      |      |         |        |      |
| A     | 4.84 | 4.18 | 4.51    | 4.55   | 0.33 |
| B     | 1.32 | 1.46 | 1.39    | 1.34   | 0.07 |
| MCN   |      |      |         |        |      |
| A     | 6.67 | 6.25 | 6.46    | 6.44   | 0.21 |
| B     | 2.03 | 2.36 | 2.20    | 2.18   | 0.17 |

The values presented in this table were generated by a least squares (LS) fit of the average experimental fitness functions plotted in Fig 1. Parameter *A* has units of μg/ml and corresponds to the cytoplasmic drug concentration, $[S]_C$, at which the normalized growth rate is half maximal. Parameter *A* can also roughly be interpreted as the apparent $K_d$ of the drug to the ribosome in addition to all other intercellular drug interactions. Parameter *B* is the Hill coefficient that describes the shape of the dose–response curve in response to the antibiotic and is thus a highly complex lumped parameter that models essentially all aspects of cellular physiology as one variable.

concentrations of antibiotic in the cytoplasm, by first modeling drug binding and unbinding to pump, and ultimately efflux into the periplasm (Fig 2A).

$$S_C + P_A \underset{k_{-1}}{\overset{k_1}{\rightleftharpoons}} P_O, \tag{2}$$

$$P_O \xrightarrow{r} S_P + P_A. \tag{3}$$

Equation (2) represents a substrate in the cytoplasm ($S_C$) binding to an available pump ($P_A$) at a rate, $k_1$, to become an occupied pump ($P_O$). The substrate in the cytoplasm can also dissociate from the pump at rate, $k_{-1}$. Once the substrate is bound to the pump, it can be transported to the periplasm ($S_P$) at pump efficiency rate, $r$, thereby returning the pump to the available state (equation 3). We can model these dynamics by taking a generalized mass action model coupled with diffusion to arrive at the following dynamic model:

$$\frac{d[S]_P}{dt} = D_1\big([S]_M - 0.5[S]_P\big) - D_2\big([S]_P - 0.5[S]_C\big) + rP_O, \tag{4}$$

$$\frac{d[S]_C}{dt} = D_2\big([S]_P - 0.5[S]_C\big) + \big(-[S]_C P_A k_1\big) + k_{-1}P_O, \tag{5}$$

$$\frac{dP_A}{dt} = -[S]_C P_A k_1 + P_O k_{-1} + rP_O, \tag{6}$$

$$\frac{dP_O}{dt} = [S]_C P_A k_1 - P_O k_{-1} - rP_O. \tag{7}$$

In equations (4) and (5), $D_1$ and $D_2$ are the diffusion constants across the outer membrane and inner membrane, respectively. The input into the model is the antibiotic concentration in the media ($[S]_M$). This is the antibiotic concentration used in the growth rate

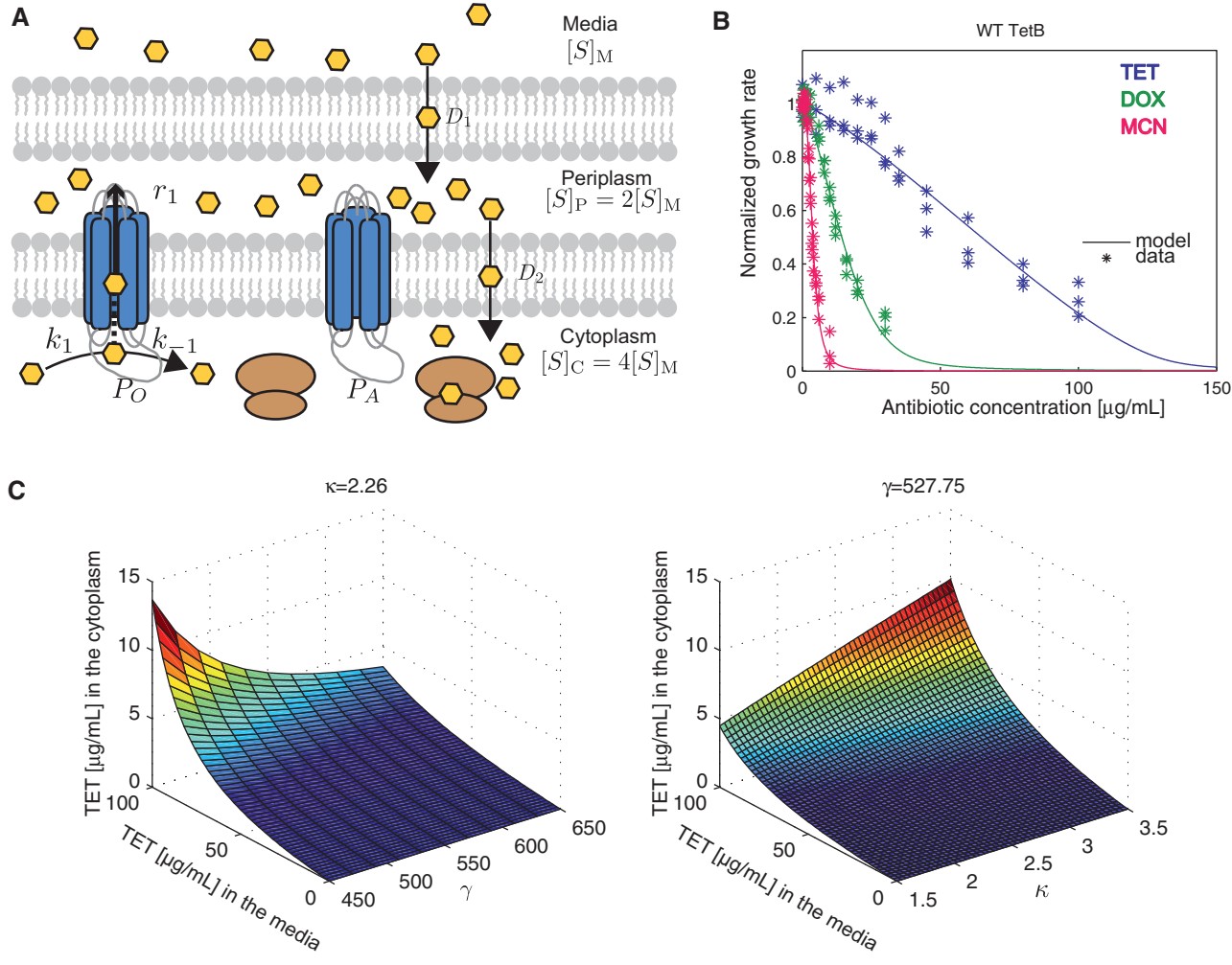

**Figure 2.  Model incorporating efflux kinetics predicts lumped physicochemical pump parameters from growth rates.**

A   TetB model description including the effect of TetB to reduce cytoplasmic concentrations of drug to relieve the inhibition of translation. $P_A$ and $P_O$ describe an available and occupied pump, respectively. Drug can bind $P_A$ by a rate, $k_1$, to create $P_O$ where two events can occur: (i) Drug disassociates from the pump at a rate, $k_{-1}$, or (ii) Drug is transported into the periplasm at a pump efficiency rate, $r$.
B   Normalized fitness functions of cells expressing wild-type *tet(B)* and their corresponding model fits in the presence of TET, DOX, and MCN. Triplicate data are shown for each antibiotic.
C   Predicted concentrations of TET in the cytoplasm as a function of TET concentration in the media, $\kappa$, and $\gamma$. The $\kappa$ and $\gamma$ values held constant in the plots are those for wild-type TetB.

Source data are available online for this figure.

assays. The ordinary differential equations then model changes in the antibiotic concentration in the periplasm ($[S]_P$) and cytoplasm ($[S]_C$) based on $[S]_M$ as well as pump kinetics. In the absence of *tet (B)* expression, we assume steady-state concentrations of tetracycline antibiotics are due solely to diffusion and the Gibbs–Donnan effect. Tetracycline is concentrated 2× from the media to the periplasm and again 2× from the periplasm into the cytoplasm resulting in a total 4× concentration in the cytoplasm, $[S]_C$, compared to the media, $[S]_M$ (Thanassi *et al*, 1995).

We then solve for the steady state of the system above. From equation (6)

$$rP_O + P_O k_{-1} = [S]_C P_A k_1. \tag{8}$$

We substitute $P_O = P_{tot} - P_A$ and solve for $P_A$

$$P_A = \frac{\kappa(P_{tot})}{\kappa + [S]_C}, \tag{9}$$

where

$$\kappa = \frac{(r + k_{-1})}{k_1}. \tag{10}$$

Furthermore, from equation (4) we have

$$D_1\left([S]_M - 0.5[S]_P\right) - D_2\left([S]_P - 0.5[S]_C\right) = -rP_O, \tag{11}$$

$$D_1\big([S]_M - 0.5[S]_P\big) - D_2\big([S]_P - 0.5[S]_C\big) = -rP_{tot} + rP_A, \tag{12}$$

$$D_1\big([S]_M - 0.5[S]_P\big) - D_2\big([S]_P - 0.5[S]_C\big) = -r\frac{P_{tot}[S]_C}{\kappa + [S]_C}. \tag{13}$$

From equation (5), we have

$$D_2\big([S]_P - 0.5[S]_C\big) + k_{-1}P_O = [S]_C P_A k_1, \tag{14}$$

$$D_2\big([S]_P - 0.5[S]_C\big) + k_{-1}\frac{P_{tot}[S]_C}{\kappa + [S]_C} = [S]_C \frac{\kappa(P_{tot})}{\kappa + [S]_C}k_1, \tag{15}$$

and rearranging terms, we get

$$D_2\big([S]_P - 0.5[S]_C\big) = \frac{P_{tot}[S]_C}{\kappa + [S]_C}(\kappa k_1 - k_{-1}), \tag{16}$$

$$D_2\big([S]_P - 0.5[S]_C\big) = r\frac{P_{tot}[S]_C}{\kappa + [S]_C}. \tag{17}$$

Plugging in equation (17) into equation (13) gives

$$[S]_M = 0.5[S]_P, \tag{18}$$

relating the concentration of substrate in the periplasm ($[S]_P$) to the concentration in the media ($[S]_M$). We can solve for the concentration of substrate in the cytoplasm ($[S]_C$) as a function of concentration of substrate in the media by substituting equation (18) into equation (17)

$$4[S]_M - [S]_C = \gamma\frac{[S]_C}{\kappa + [S]_C}, \tag{19}$$

where

$$\gamma = \frac{2rP_{tot}}{D_2}. \tag{20}$$

Finally, solving for $[S]_C$ as a function of $[S]_M$ gives

$$[S]_C = \frac{-\big(\gamma + \kappa - 4[S]_M\big) \pm \sqrt{\big(\gamma + \kappa - 4[S]_M\big)^2 + 16\kappa[S]_M}}{2}, \tag{21}$$

for which the positive solution is the only feasible solution. We assume that substrate diffusion across the inner membrane, $D_2$, remains constant. In order to fit parameters using growth rate data, we combine equation (21) with equation (1), which relates growth rate to the concentration of substrate in the cytoplasm, $GR = 1 - \frac{[S]_C^B}{A^B + [S]_C^B}$. Efflux pumps counteract substrate diffusion, and therefore, we cannot know the concentration of substrate in the cytoplasm, $[S]_C$, by simply knowing the substrate concentration in the media, $[S]_M$. Using equation (1) and the global parameters $A$ and $B$ determined in the pump-free system allow us to calculate $[S]_C$ from fitness (growth rates) of strains with the transporter. Equation (21) is then used for the prediction of the pump parameters, $\gamma$, relating to the total activity of the transporter and $\kappa$, which

roughly corresponds to the affinity of substrate to the transporter, and $P_{tot}$ is the total amount of protein.

This model was used to fit fitness data of strains expressing a chromosomal copy of *tet(B)* at a wide range of TET, DOX, and MCN concentrations (Fig 2). *In vivo*, the expression of most efflux pumps is tightly regulated by repressors as their expression in non-selective environments can lead to substantial fitness costs (Nguyen *et al*, 1989). In naturally occurring biological systems, *Tet(B)* is controlled by the TetR repressor whereby expression is dynamically controlled by the concentration of tetracycline in the environment (Møller *et al*, 2016). In an effort to more easily predict pump physicochemical properties from fitness in selective environments, we removed these regulatory dynamics by TetR. Here, *tet(B)* is expressed by the arabinose-inducible, glucose-repressible promoter, pBAD, where we use a saturating arabinose concentration to obtain a largely homogeneous cellular population expressing *tet(B)* (Khlebnikov *et al*, 2000). We ensured moderate translational expression of TetB by engineering a ribosome binding site (RBS) sequence that provided measurable resistance at minimal fitness cost (Materials and Methods; Salis *et al*, 2009). The final construct was integrated into the *E. coli* BW25113 chromosome using a site-specific method for insertion at the Tn7 attachment sites downstream of the highly conserved glutamine synthetase gene, *glmS* (McKenzie & Craig, 2006). Our host strain BW25113 has a deletion in the genes necessary for arabinose catabolism; therefore, *tet(B)* expression is under constant induction in the presence of saturating arabinose, and we assume that $P_{tot}$ is constant within our system (Grenier *et al*, 2014). Previous work has shown that inadvertent coupling between growth rate and gene expression can lead to bistability in growth response of a drug-resistant *E. coli* strain in the presence of chloramphenicol (Deris *et al*, 2013). The authors show this using a phenomenological model, where concentrations of the drug-deactivating enzyme chloramphenicol acetyltransferase are growth rate dependent. Here, we greatly minimize any measurable coupling between expression of TetB and growth rate of the cell through the constant saturating induction of the inducible promoter. This allows us to create a simpler physiological model that can provide more information about the kinetic properties of the protein.

Using our physicochemical-fitness model, we generated least squares fits to normalized fitness functions for cells expressing *tet (B)* under selection by TET, DOX, or MCN and determined parameters $\kappa$ and $\gamma$ (Fig 2B, Appendix Table S1 and Code EV2). In our system, we assume that $P_{tot}$ and drug diffusion across the inner membrane ($D_2$) remain constant. Here, TetB more readily pumps out TET compared to DOX and MCN, which is consistent with previous studies performed with *tet(B)* (Testa *et al*, 1993; Nguyen *et al*, 2014). Figure 2C shows how the final steady-state concentration of TET in the cytoplasm relates to a given concentration in the media based on equation (8). We find that there is a strong correlation between $\gamma$ and changes in growth rate. This is expected since $\gamma$, equation (20), is proportional to protein concentration and pumping efficiency, which greatly impacts the cytoplasmic concentration of antibiotic. Similar to our previous report, we find that $\kappa$, relating to substrate affinity defined in equation (10), is sensitive to the initial plateau of the fitness function at lower concentrations of antibiotic while $\gamma$, relating to total activity, is sensitive to the changes in the

growth rate as substrate concentration increases (Appendix Fig S1; Walkiewicz *et al*, 2012).

### Fitness functions of *tet(B)* variants can be modeled accurately to reveal relevant physicochemical properties of efflux pumps

To test the ability of our physicochemical-fitness model to determine physicochemical parameters relating to substrate affinity and total protein activity from fitness, we tested known TetB variants that have been shown to reduce TetB function (Sapunaric & Levy, 2005). Variants were also selected for their predicted location in the conserved MFS fold for TetB and their potential to alter pumping efficiency and substrate binding affinity. The majority of MFS transporters have an N- and C-terminal domain, each consisting of six transmembrane helices and joined by the cytoplasmic interdomain loop (Fig 3A).

There are three groups of four alpha helices that interact to form the canonical MFS fold. Here, we use the nomenclature from Quistgaard *et al*, 2016 for the helices. A-helices shown in blue (TM1, TM4, TM7, and TM10) are positioned at the center of the transporter and make up the transport path needed for substrate binding and exchange or cotransport coupling. B-helices shown in green (TM2, TM5, TM8, and TM11) are positioned outside of the core helices and have been shown to help mediate the N- and C-domain interface and may also participate in substrate binding. C-helices shown in gray (TM3, TM6, TM9, and TM12) are positioned on the outside of the core helices and have been postulated to primarily contribute more to the structure than the function of the transporter (Yan, 2013).

In addition, we tested four variants within the interdomain loop of TetB at residues D190 and E192 that have been suggested to be important for substrate specificity and reduce tetracycline sensitivity

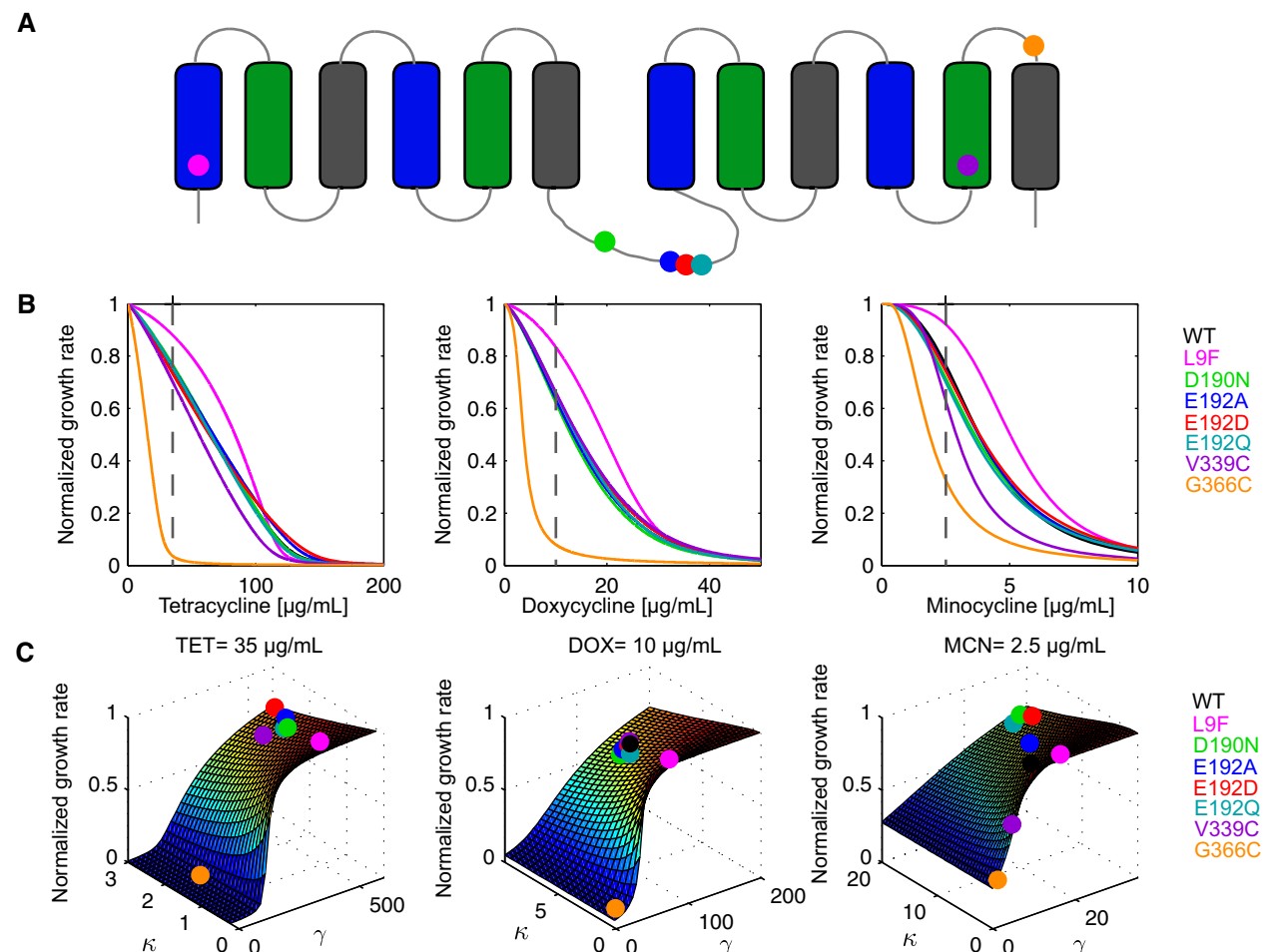

**Figure 3. Physicochemical properties of TetB variants can be determined from cellular fitness.**

A  Variant locations on the predicted TetB topology. A-helices are shown in blue, B-helices are shown in green, and C-helices are shown in gray.

B  Model fits for *Escherichia coli* BW25113 strains expressing a chromosomal copy of *tet(B)^WT^* and seven variants in TET, DOX, and MCN. Our model fit triplicate fitness function data for all strains (Appendix Fig S2). At low drug concentrations for all antibiotics, *tet(B)^G366C^* exhibits much slower growth rates than wild type. Compared to wild type, strains expressing *tet(B)^L9F^* exhibit similar growth rates at low and high drug concentrations but have faster growth rates at intermediate drug concentrations.

C  Fitness landscapes with mutants overlaid for TET = 35 µg/ml, DOX = 10 µg/ml, and MCN = 2.5 µg/ml corresponding to the dashed lines in panel (B).

Source data are available online for this figure.

(Tamura *et al*, 2001; Sapunaric & Levy, 2005). Studies with other MFS transporters, however, indicate that the interdomain loop is necessary for movement and flexibility between the N- and C-terminal domains to facilitate the pumping mechanism required for substrate translocation (Law *et al*, 2008; Dang *et al*, 2010).

In total, seven different *tet(B)* variant constructs were created using site-directed mutagenesis and integrated into the chromosome of the parent strain for subsequent growth rate analysis in TET, DOX, and MCN (Fig 3A and B, and Appendix Fig S2). Insertion and expression of *tet(B)* or variants resulted in modest fitness costs within our cellular system (Appendix Fig S3). For all tested variants, an increase in drug concentration caused a decrease in growth rate. Surprisingly, interdomain loop variants $tet(B)^{D190N}$, $tet(B)^{E192D}$, and $tet(B)^{E192Q}$ identified by Sapunaric and Levy (2005) as reducing TetB function showed fitness nearly identical to cells expressing $tet(B)^{WT}$ and as expected, our model correctly predicted similar $\kappa$ and $\gamma$ parameters as can be seen by their overlap in the fitness landscape in Fig 3C. Cells expressing $tet(B)^{V339C}$, located in B-helix TM11, had a greater decrease in fitness compared to cells expressing $tet(B)^{WT}$ in TET and MCN. In DOX, both fitness functions overlapped showing that a decrease in fitness is not observed during DOX selection. In Fig 3C, parameters for TetB$^{V339C}$ can be differentiated from TetB$^{WT}$ in TET and MCN but is clustered with TetB$^{WT}$ in DOX. The strain expressing $tet(B)^{L9F}$ located in A-helix TM1, surprisingly, is noticeably more fit than cells expressing $tet(B)^{WT}$ in all tested drugs although in the work of Sapunaric and Levy this variant was shown to produce a decrease in resistance. Interestingly, $\gamma$ remained similar to WT, while $\kappa$ changed 2.5-fold in all drugs (Appendix Table S1). This result echoes our previous work showing that small changes in protein function can result in large fitness effects (Walkiewicz *et al*,

2012). Cells expressing $tet(B)^{G366C}$, located in C-helix TM12, were significantly less fit than cells expressing $tet(B)^{WT}$. This mutation resulted in severe decreases in both $\kappa$ and $\gamma$ in all drugs.

## Small changes in *in vivo* protein levels play a decidedly important role in determining fitness

The total activity of a transporter is a result of both its pump efficiency and concentration. In the absence of knowing protein concentration, it is impossible to obtain information regarding changes in pumping efficiency from the lumped parameter, $\gamma$, determined solely from growth rates (equation 9). Although we used a collection of single amino acid substitutions, it is likely that some of these mutations may have affected the steady-state concentration of TetB, and thus, it would be inaccurate to ascribe changes in $\gamma$ solely to changes in pumping rate. We therefore performed Western blot analysis to assess the potential importance for small changes in protein concentration to drive increased fitness. We used an antibody toward TetB to quantify relative total protein concentration using the membrane fraction of strains expressing chromosomal copies of $tet(B)^{WT}$ and variants harvested at mid-exponential growth phase (Appendix Figs S5 and S6). As shown in Fig 4, the *in vivo* protein level of TetB$^{L9F}$ is ~50% higher than wild type, while TetB$^{V339C}$ and TetB$^{G366C}$ appear somewhat lower by ~20 and ~50%, respectively. The *in vivo* expression of all interdomain loop variants was similar to wild type and, when combined with their fitness data in Fig 3B, indicates that these variants do not alter TetB physicochemical properties in our system. Subsequent analyses of these variants are discussed in the Appendix Figs S7 and S8. These data highlight the importance of incorporating *in vivo* protein levels in the physicochemical-fitness model as this factor played an important

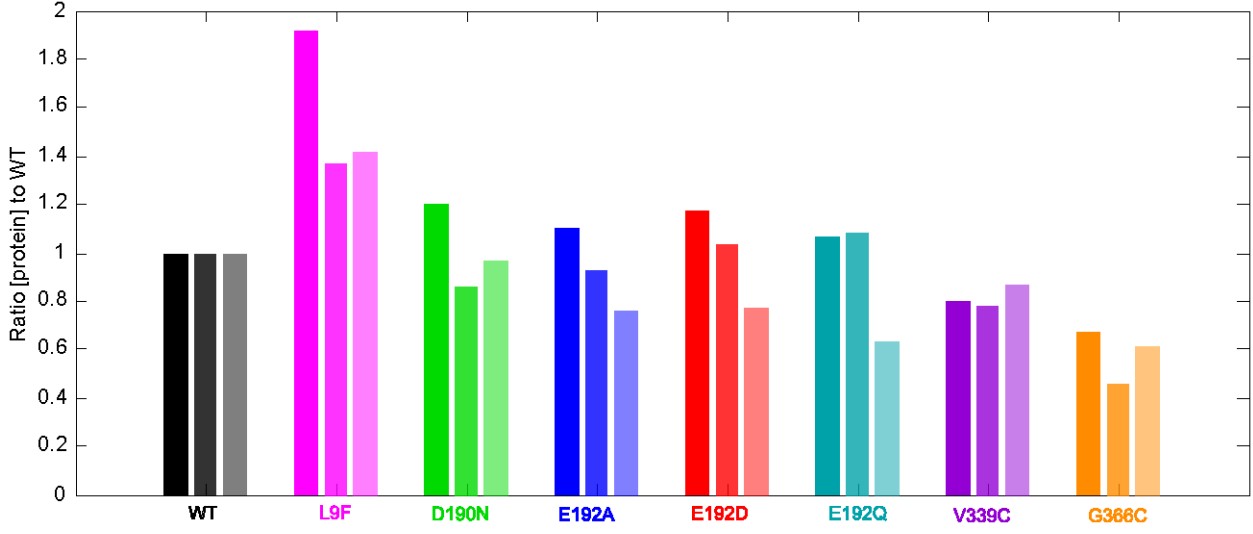

**Figure 4.    *In vivo* protein levels of TetB variants relative to wild type determined by Western blot analysis of triplicate cell membrane fractions using an antibody for the C-terminal tail of TetB.**

*Escherichia coli* BW25113 strains expressing a chromosomal copy of $tet(B)^{WT}$ or variant were grown in 2% arabinose-supplemented LB in the absence of drug and harvested at mid-exponential phase. Membrane samples were prepared from three individual colonies grown independently (Appendix Figs S5 and S6). Compared to wild type, strains expressing $tet(B)^{L9F}$ consistently have higher *in vivo* protein levels whereas strains expressing $tet(B)^{V339C}$ and $tet(B)^{G366C}$ have lower *in vivo* protein levels as detected by Western blot analysis. All four variants located in the interdomain loop exhibited *in vivo* protein levels similar to wild type.

Source data are available online for this figure.

role in the increased fitness of cells expressing *tet(B)$^{L9F}$* and decreased fitness of cells expressing *tet(B)$^{G366C}$*.

### The relative change in substrate binding and pumping efficiency rates can be determined after incorporation of protein concentration into the physicochemical-fitness model

Integrating *in vivo* protein levels into our physicochemical-fitness model allows for the uncoupling of lumped parameters $\gamma$ and $\kappa$ into the specific properties describing substrate binding rate, $k_1$, and pumping efficiency rate, $r$. Although it is not possible to find explicit values for the rate constants, we can determine the effective relative changes in $k_1$ and $r$ for each variant across TET, DOX, and MCN. Relative changes in $r$ were determined by incorporating protein levels directly into our definition of $\gamma$ in equation (9). We determined relative changes in $k_1$ through the ratio $\gamma/\kappa$. Consider

$$\frac{\gamma}{\kappa} = \frac{rk_1}{(r+k_{-1})} \cdot \frac{P_{tot}}{D_2}. \tag{22}$$

If we divide through by $k_1$, we get

$$\frac{\gamma}{\kappa} = \frac{r}{(r/k_1 + \varepsilon)} \cdot \frac{P_{tot}}{D_2}, \tag{23}$$

where $\varepsilon = k_{-1}/k_1$. We assume that the off binding rate of substrate to TetB is much lower than the on binding rate (i.e., $\varepsilon = k_{-1}/$

$k_1 \ll 1$). Expanding equation (23) around $\varepsilon$ in a Taylor series to second order, we get

$$\frac{\gamma}{\kappa} \approx \left( k_1 - \frac{k_1^2}{r}\varepsilon + \frac{k_1^3}{r^2}\varepsilon^2 + \dots \right) \cdot \frac{P_{tot}}{D_2}. \tag{24}$$

Substituting back in the definition for $\varepsilon$, we find that if $k_{-1} \ll r$, then a zero-order approximation holds and we should expect that $\kappa = (r + k_{-1})/k_1$ correlates linearly with $\gamma = 2rP_{tot}/D_2$. In this case, the relationship is described by the zero-order approximation

$$\frac{\gamma}{\kappa} \approx k_1 \frac{2P_{tot}}{D_2}. \tag{25}$$

This assumption was tested for each *tet(B)* variant. We took a Monte Carlo approach to generate 1,000 samples of fits on $\kappa$ and $\gamma$ through a random selection of global parameters $A$ and $B$ from the estimated probability density functions (details in Materials and Methods). Figure 5A shows the fits for strains expressing *tet(B)$^{WT}$* and variants *tet(B)$^{L9F}$*, *tet(B)$^{V339C}$*, and *tet(B)$^{G366C}$* where, indeed, there appears to be a linear correlation between $\kappa$ and $\gamma$. The calculated ratio $\gamma/\kappa$ for these strains are shown in Fig 5B where error bars correspond to error from $A$ and $B$ parameter estimations. The majority of the cases have relatively modest error indicating a consistent measurement of ratio $\gamma/\kappa$ even in the presence of uncertainty in $A$ and $B$. The large error within ratio $\gamma/\kappa$ for both TetB$^{V339C}$ and

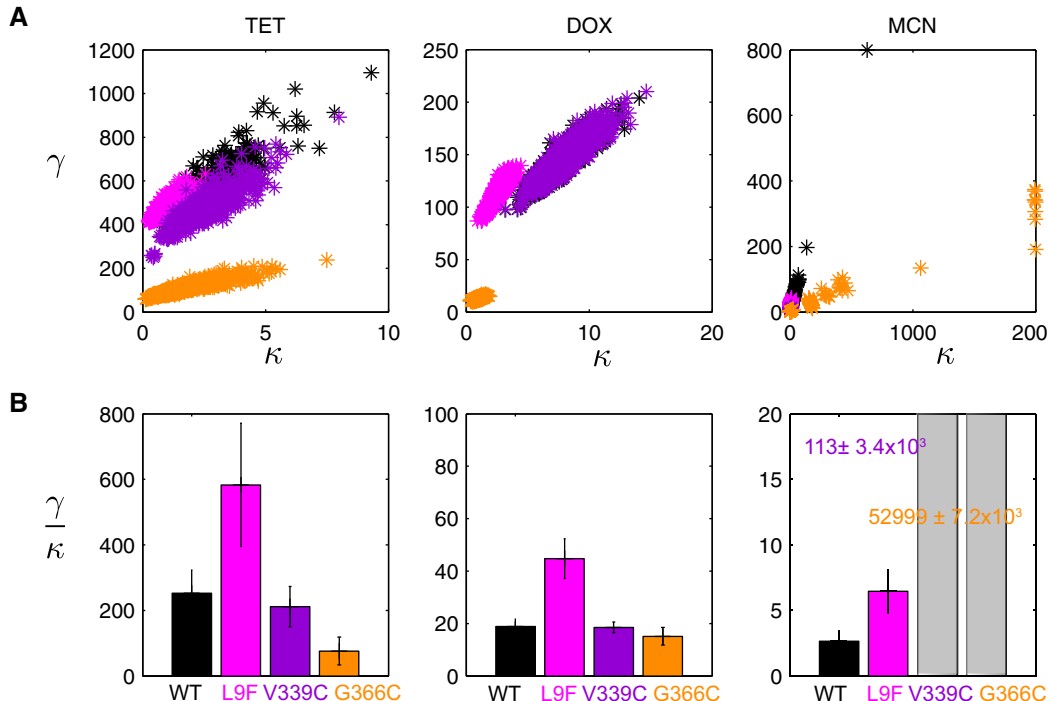

**Figure 5. Statistics of ratio $\gamma/\kappa$ allow us to test first-order approximation assumptions.**

A   The spread of $\kappa$ and $\gamma$ fits for 1,000 randomly selected parameters $A$ and $B$ from a set gamma distribution shows a linear correlation between $\kappa$ and $\gamma$.

B   The calculated mean and error of $\gamma/\kappa$ for the sampled parameter fittings shown in Appendix Fig S2. The large error seen for TetB$^{V339C}$ and TetB$^{G366C}$ in MCN reveals that we cannot make the first-order approximation within $\frac{\gamma}{\kappa} \approx k_1 \frac{2P_{tot}}{D_2}$.

TetB$^{G366C}$ in MCN informs us that we cannot make any conclusions about fold changes in the parameter $k_1$ with confidence. This large error is due to the large uncertainty in the $\kappa$ prediction for these two variants, which amplifies significantly in the ratio $\gamma/\kappa$ (Appendix Table S2). We hypothesize that the large error in $\kappa$ is due to the nearly negligible fitness increase produced by strains expressing $tet(B)^{V339C}$ and $tet(B)^{G366C}$ compared to the strain without a $tet$ $(B)$ allele showing an insensitivity of our model to confidently predict $\kappa$ (Appendix Fig S9) at very low levels of protein activity. In all further analysis, we omitted any fold change calculations in $k_1$ for TetB$^{V339C}$ and TetB$^{G366C}$ in MCN; however, we do obtain a confident prediction of $\gamma$ for TetB$^{V339C}$, which relates closely to the drop-off point of the fitness function rather than the subtle difference in the slope.

Next, we calculate relative changes in parameters $k_1$ and $r$ incorporating uncertainty in $A$ and $B$ as well as protein concentration measurements. Although we were able to determine relative protein levels within 30%, the challenge in precisely estimating *in vivo* protein levels for membrane proteins is a large cause of uncertainty in our calculations. Therefore, the error propagated here is hindered by the resolution of the $P_{tot}$ estimation (Fig 4). Additionally, the uncertainty in our least squares fit of $\kappa$ and $\gamma$ is from error propagated from the estimation of $A$ and $B$. In order to calculate the terms involving $\kappa$ and $\gamma$ for the mean and variance calculations of the relative changes in $k_1$ and $r$, we used the data generated from the Monte Carlo approach in Table 1.

Incorporating relative *in vivo* protein levels and uncertainty in $A$ and $B$, we used equation (25) to determine relative change in substrate binding rate, $k_1$, and $\gamma$ defined in equation (20) to determine the relative change in pumping efficiency, $r$. The relative change in the parameter $r$ is calculated as

$$\Delta r = \frac{r_m - r_{wt}}{r_{wt}} \\ = \frac{\gamma_m}{\gamma_{wt}} \frac{P_{tot,wt}}{P_{tot,m}} - 1 . \tag{26}$$

This is found by solving for $r$ in equation (20), which defines $\gamma = 2rP_{tot}/D_2$. We apply the expectation operator and get that the mean of $\Delta$r is

$$\mathbf{E}[\Delta r] = \mathbf{E}\left[\frac{\gamma_m}{\gamma_{wt}}\right] \mathbf{E}\left[\frac{P_{tot,wt}}{P_{tot,m}}\right] - 1, \tag{27}$$

and the corresponding variance is

$$\mathrm{Var}(\Delta r) = \mathbf{E}[\Delta r^2] - (\mathbf{E}[\Delta r])^2 \\ = \mathbf{E}\left[\left(\frac{\gamma_m}{\gamma_{wt}}\right)^2\right] \mathbf{E}\left[\left(\frac{P_{tot,wt}}{P_{tot,m}}\right)^2\right] - \left(\mathbf{E}\left[\frac{\gamma_m}{\gamma_{wt}}\right] \mathbf{E}\left[\frac{P_{tot,wt}}{P_{tot,m}}\right]\right)^2 . \tag{28}$$

Similarly, with the expression $\gamma/\kappa \approx k_1 \; P_{tot}/D_2$ and equation (10), we have

$$\Delta k_1 = \frac{\gamma_m \cdot \kappa_{wt}}{\gamma_{wt} \cdot \kappa_m} \frac{P_{tot,wt}}{P_{tot,m}} - 1 \tag{29}$$

with mean

$$\mathbf{E}[\Delta k_1] = \mathbf{E}\left[\frac{\gamma_m \cdot \kappa_{wt}}{\gamma_{wt} \cdot \kappa_m}\right] \mathbf{E}\left[\frac{P_{tot,wt}}{P_{tot,m}}\right] - 1 \tag{30}$$

and variance

$$\mathrm{Var}(\Delta k_1) = \mathbf{E}[\Delta k_1^2] - (\mathbf{E}[\Delta k_1])^2 \\ = \mathbf{E}\left[\left(\frac{\gamma_m \cdot \kappa_{wt}}{\gamma_{wt} \cdot \kappa_m}\right)^2\right] \mathbf{E}\left[\left(\frac{P_{tot,wt}}{P_{tot,m}}\right)^2\right] - \left(\mathbf{E}\left[\frac{\gamma_m \cdot \kappa_{wt}}{\gamma_{wt} \cdot \kappa_m}\right] \mathbf{E}\left[\frac{P_{tot,wt}}{P_{tot,m}}\right]\right)^2 . \tag{31}$$

Figure 6A shows the relative changes in $k_1$ and $r$ across the variants compared to TetB$^{WT}$. TetB$^{G366C}$ significantly reduces the relative pumping efficiency but minimally impacts the binding rate constant in TET and DOX. TetB$^{L9F}$ produces both an increase in relative binding affinity and decreases in pumping efficiency across all drugs by comparable magnitudes. We know from Fig 2C that the lumped parameter $\gamma$ is primarily implicated in altering the cytoplasmic drug concentration. For cells expressing $tet(B)^{L9F}$, their decrease in pumping efficiency is compensated for by the increase in protein levels. The effect of mutant TetB$^{V339C}$, interestingly, is quite variable across the three different drugs even when Appendix Fig S4 shows a consistent change in drug specificity across all mutants. Although the relative pumping efficiency is slightly larger by ~25%, the *in vivo* protein level of TetB$^{V339C}$ is slightly lower than WT resulting in overlapping fitness functions in DOX.

### The physicochemical-fitness model can be used to screen a large plasmid-encoded protein library

Having established that chromosomally encoded variants of TetB could be well modeled by our approach, we went on to test the efficacy of our method to a plasmid-encoded library of TetB variants. Plasmid-encoded libraries of protein variants are frequently used for high-throughput screening and, therefore, it was important to establish the utility of the approach to a format used more broadly during drug development. The lumped parameters $\kappa$ and $\gamma$ are determined solely from growth rates and provide a way for libraries to be screened for a desired trait, such as altered substrate affinity and/or total activity. To demonstrate the library screening potential of our model, we created an error-prone PCR mutant $tet(B)$ library harbored on a low-copy-number plasmid and performed growth analysis on a small subset of variants. In addition, new $A$ and $B$ global parameters describing the baseline response of this system to TET were determined using a strain containing an empty plasmid. These values were calculated to be $A = 1.91$ and $B = 1.42$ and are comparable to the first analysis performed in the absence of empty plasmid (Table 1 and Dataset EV2). Functional TetB variants were isolated and initially screened via replica plating and growth analysis at a single TET concentration (details in Materials and Methods). We analyzed seven TetB variants selected from an initial functional screen and performed growth analysis in TET. Of the seven variants, five were able to grow in the TET concentrations tested in the growth assay. Two variants exhibited fitness functions similar to WT, one exhibited highly variable fitness functions, and two

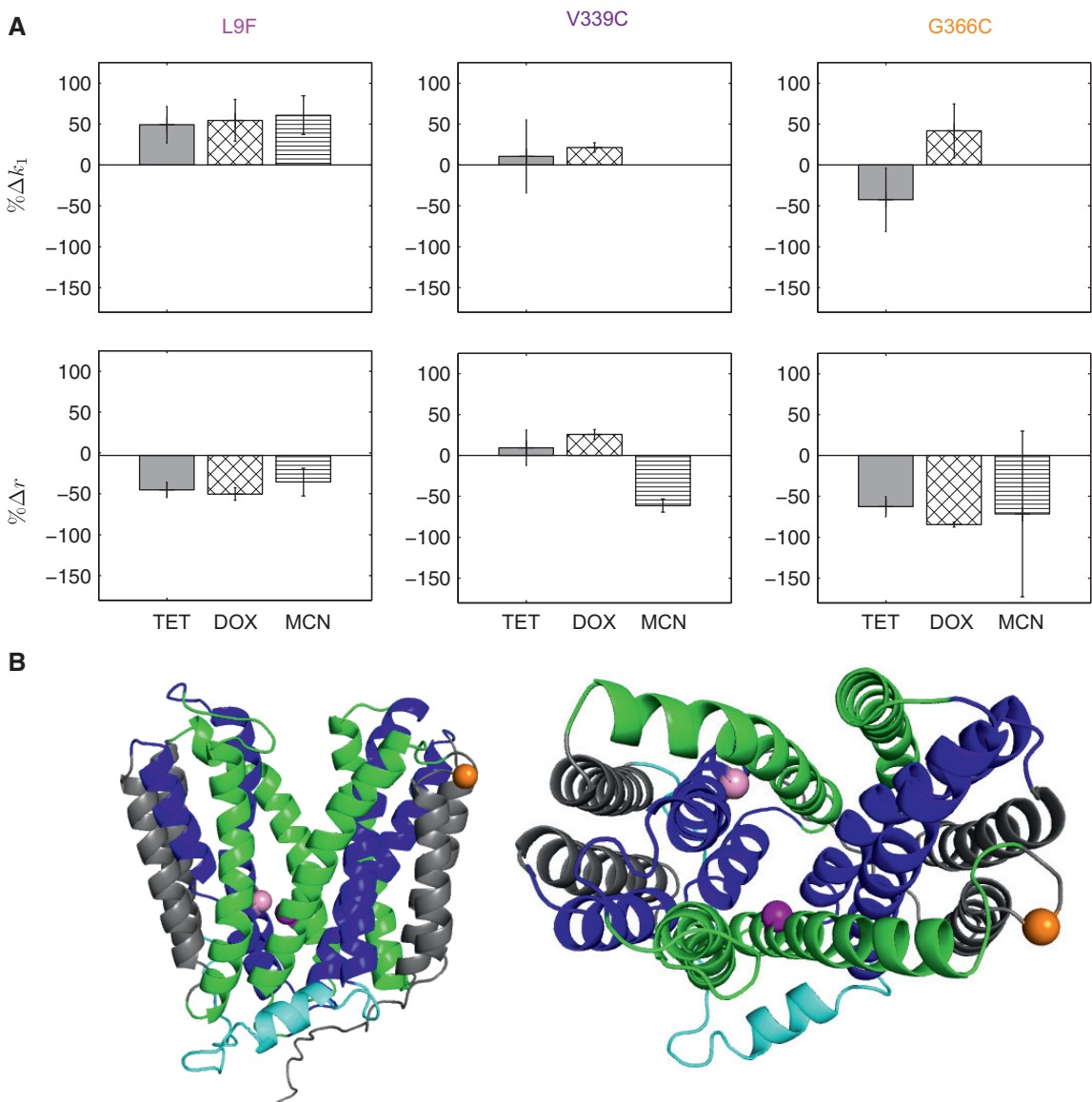

**Figure 6. Relative changes in $k_1$ and $r$ can be uncoupled from $\kappa$ and $\gamma$ using *in vivo* protein levels.**

A   Changes in drug binding affinity $\Delta k_1$ and pumping efficiency $\Delta r$ for TetB variants relative to wild type in TET, DOX, and MCN. Error bars are comprised from propagation of errors from $A$ and $B$ global parameter modeling and *in vivo* protein-level determination.

B   Homology model of TetB using YajR (PDB 3wdo) built using Phyre2 where the periplasmic side is on top and the cytoplasmic side is on the bottom. The A-helices are shown in blue, B-helices are shown in green, C-helices are shown in gray, and the interdomain loop is in cyan. Spheres indicating variant positions were made using PyMOL (L9, pink; V339, purple; G366, orange). Right image depicts the view through the central TetB pore from the periplasm into the cytoplasm.

variants, variant 6 and variant 7, were less fit than WT (Fig 7A, Appendix Fig S10 and Table S3, Appendix Source Data).

Variant 6 contained three SNPs; K183E is located within the interdomain loop, A290V is located in C-helix TM 9, and T326I is located within the short cytoplasmic loop connecting TM 10 and TM 11 (Fig 7C). The mutations within this variant produce a 187% increase in $\kappa$ and 86% increase in $\gamma$ relative to WT (Fig 7B). The increase in $\gamma$ represents an increase in total protein activity of variant 6; however, there is an even greater increase in $\kappa$ representing decreased binding affinity to TET resulting in an overall decrease in fitness compared to WT (Fig 7A and B). Variant 7 also has three

SNPs: F151I located in B-helix TM 5, F179L located within the interdomain loop, and T326A, interestingly, at the same locus as variant 6 mutation T326I. In the strain expressing plasmid variant 7, we observe a 30% decrease in the $\gamma$ parameter while the $\kappa$ remains unchanged compared to WT TetB in plasmid (Fig 7B).

## Discussion

Determination of the physicochemical properties of a single specific protein solely from fitness poses an interesting but difficult challenge

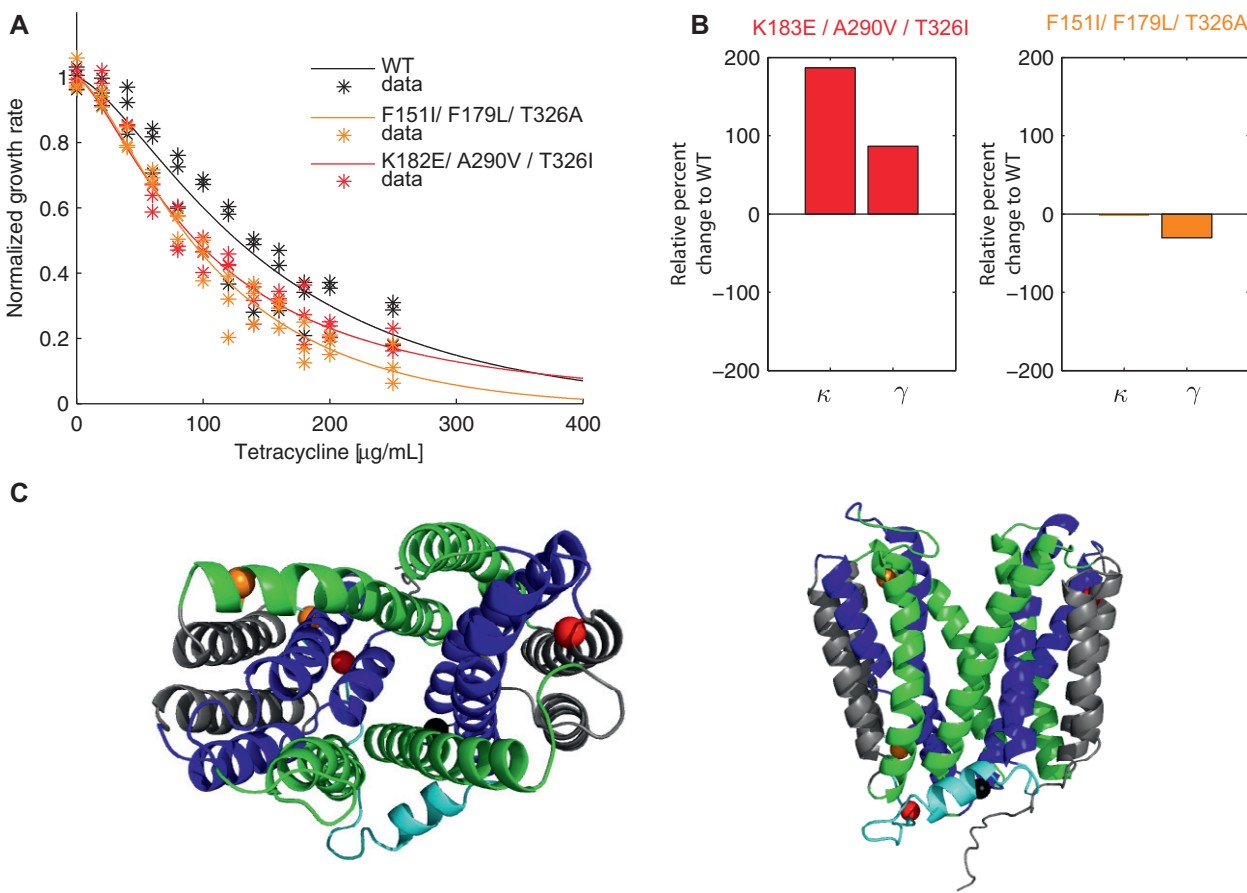

**Figure 7.  The physicochemical-fitness model can be used to screen plasmid-encoded libraries of protein variants.**

A  Triplicate normalized fitness functions (*) for *Escherichia coli* BW25113 strains harboring plasmid copies of $tet(B)^{WT}$, $tet(B)^{K183E,A290V,T326I}$ (variant 6) and $tet(B)^{F151I,F179L,T326A}$ (variant 7) in TET and their corresponding least squares model fits (solid line) where both variants display decreased fitness in TET compared to WT.

B  Percent change relative to WT in parameters $\kappa$ and $\gamma$ for strains expressing $tet(B)^{K183E,A290V,T326I}$ and $tet(B)^{F151I,F179L,T326A}$. Parameter values were determined by the least squares model fits in (A).

C  Homology model of TetB using YajR (PDB 3wdo) built using Phyre2 where the periplasmic side is on top and the cytoplasmic side is on the bottom. The A-helices are shown in blue, B-helices are shown in green, C-helices are shown in gray, and the interdomain loop is cyan. Spheres indicating variant positions were made using PyMOL (K183 and A290, red; F151 and F179, orange; T326, black, as this residue is altered in variants 6 and 7). The right image depicts the view through central TetB pore from the periplasm into the cytoplasm.

as the phenotype of an organism is the summation of the physico-chemical properties of all cellular proteins and their interactions (Deris *et al*, 2013). Here, we take an otherwise complex phenotype, resistance to tetracycline mediated by TetB, and examined the fitness of *E. coli* at varying environmental conditions to link *in vivo* TetB function to its physicochemical properties. Wild-type *tet(B)* and strategically chosen variants were used to test the hypothesis that we could extract meaningful information regarding substrate binding and pumping efficiency solely from growth at selective conditions and relative *in vivo* protein concentrations. We built a simple but robust mathematical model that incorporates substrate diffusion, cytoplasmic substrate concentration, and growth at selective conditions to predict relative changes in these pump parameters and could be extended as a general framework for other MFS family members. We further expanded upon this framework to show that our model can be used to screen and assay protein libraries solely from growth at selective conditions to help identify variants of a desired trait.

We first determined the global parameters *A* and *B* that define our cellular system from growth analysis of our host cell without *tet (B)* in the chromosome using a Hill-function relating growth rate inhibition and cytoplasmic drug concentration. Interestingly, our host cell has an intrinsically higher resistance to semisynthetic, second-generation tetracycline antibiotics DOX and MCN compared to TET. This unexpected feature emphasizes the importance of obtaining *A* and *B* for all tested substrates as the determination of the relative pump parameters is dependent on these initial global parameters.

From our steady-state model, we find that the cellular fitness response emerges from a balance between diffusion and transport, which takes on dynamics analogous to Michaelis–Menten kinetics typically used to describe transporters (Law *et al*, 2007; Sun *et al*, 2012). Our model was able to extrapolate information from the fitness function curves and estimate two lumped parameters, $\kappa$ and $\gamma$. $\kappa$ is the ratio between the reaction rates resulting in unbound and

bound states of the pump while $\gamma$ is proportional to the pump efficiency and protein concentration. These parameters are loosely analogous to $K_m$ and $V_{\mathrm{max}}$ in Michaelis–Menten kinetics, respectively. Our model cannot distinguish between changes in pumping efficiency or protein levels as both equally impact $\gamma$, the apparent $V_{\mathrm{max}}$. However, by incorporating relative *in vivo* protein concentration, we were able to use the model to extrapolate more detailed information regarding specific protein activities. It is important to note that the model does not require measurement of protein concentration *in vivo* but only aids in discerning between changes in catalytic activity ($k_{\mathrm{cat}}$) versus total protein concentration ($P_{tot}$).

Although absolute values of $\kappa$ and $\gamma$ were shown to be sensitive to predicted values of $A$ and $B$, we note that as a first screening process; Fig 5A still demonstrates a visible distinction for variants whose fitness differs from wild type. These parameters can serve as a good initial screening method when looking for variants with specific qualities. Furthermore, comparing Figs 5A with 6A highlights the sensitivity of the model. A main source of error in this analysis arises from the estimation of protein levels relative to WT. This additional analysis, however, allowed for better understanding of the specific physicochemical property impacted by these variants as it relates to the overall performance of TetB.

Of the seven chromosomally encoded variants tested in this study, three conferred significant changes in fitness allowing for their relative changes in substrate binding rate, $k_1$, and pumping efficiency, $r$, to be determined. TetB$^{L9F}$ is located in the A-helices postulated to make up the transport path and be involved in substrate binding (Fig 6B). Modeling growth rate information and incorporating protein levels revealed that, indeed, this mutation impacted binding rate; however, pumping efficiency was also impacted to a comparable level in all drugs. Despite the decreased pumping efficiency, the increase in fitness experienced by the strain expressing *tet(B)*$^{L9F}$ is due to a combination of both greater protein levels and an increase in $k_1$ relative to cells expressing *tet(B)*$^{WT}$. TetB$^{G366C}$ is located in the C-helices that are proposed to be involved in maintaining the structural integrity of MFS transporter (Yan, 2013). The strain expressing *tet(B)*$^{G366C}$ had a severe decrease in fitness compared to wild type in all drugs. Incorporation of *in vivo* protein levels into modeling revealed that, indeed, this mutation impacts pumping efficiency to a greater extent than substrate binding consistent with our current knowledge of the C-helices (Fig 6A and B). The decrease in fitness of cells expressing this variant is due to a combination of lower *in vivo* protein levels and pumping efficiency relative to the *tet(B)*$^{WT}$ strain. Cells expressing B-helix mutant *tet(B)*$^{V339C}$ showed minimal but statistically significant increases in $k_1$ in DOX and no change in TET and could not be resolved for MCN. Interestingly, the strain expressing *tet(B)*$^{V339C}$ manifested changes in pumping efficiency, $r$, for each drug, but the extent of change was dissimilar for all drugs, suggesting that while there was no change in specificity, there were generalized changes in how well this variant could pump across the membrane (Appendix Fig S4). Since the B-helices are postulated to mediate the N- and C-domain interface, it is possible that the mutation V339C is impacting this interface in such a way in TetB that substrate transport is differentially altered in TET, DOX, and MCN.

We tested four variants located in the interdomain loop: TetB$^{D190N}$, TetB$^{E192A}$, TetB$^{E192D}$, and TetB$^{E192Q}$. These variants did not significantly alter cellular fitness in our study, differing from a previous study, which suggested that these loop residues contribute to substrate specificity (Sapunaric & Levy, 2005). One reason for this difference may lie in the differing cellular context for which the studies were performed. Previous reports have shown that *tetB* expression is dynamically regulated by the TetR repressor, in direct response to tetracycline concentration while in our system, TetR is absent and the *in vivo* TetB concentration is kept constant (Møller *et al*, 2016).

We also show that our system can be used to screen and assay protein libraries by creating and testing a *tet(B)* plasmid mutant library in TET. We quickly created, screened, and determined the lumped parameter $\kappa$ and $\gamma$ values for *tet(B)* library variants using only whole cells and without the challenging task of protein purification. The modeling of the strain harboring plasmid variant six triple mutant *tet(B)*$^{K183E,A290V,T326I}$ exhibited weaker substrate binding for TET as indicated by a 187% increase in $\kappa$, the apparent $K_m$, but an increase in total protein activity indicated by a 86% increase in the $\gamma$ parameter. We know from Fig 2C that $\gamma$ has a larger influence on the concentration of TET in the cytoplasm than $\kappa$ and, therefore, more heavily influences cellular fitness; however, the magnitude of the decrease in $\kappa$ is ~2 fold more than the increase in $\gamma$ resulting in a modest fitness decrease compared to WT. Plasmid variant 7 triple mutant *tet(B)*$^{F151I,F179L,T326A}$ solely exhibited a 30% decrease in total protein activity. Two of the three mutations within variant 7 alter bulky aromatic phenylalanine residue to smaller aliphatic isoleucine and leucine residues, which may impact the overall ability of TetB to pump out TET. Interestingly, both variants 6 and 7 harbor mutations, which alter polar residue T326 to aliphatic residues. Without analyzing the specific single mutations of these triple mutants, we can only determine how the combined effects of these mutations impact protein parameters.

Although our characterization method is able to provide meaningful information about the mutated residue and location, our simplified model does not incorporate all possible kinetic events needed for substrate efflux or drug:cellular interactions. For example, our model does not incorporate the details about co-substrate transport for H$^+$ in our TetB system or how cytoplasmic drug binding to the ribosome may impact the concentration of $[S]_C$ available for TetB efflux. Additionally, we have omitted the binding of cytoplasmic drug $[S]_C$ to a ribosome. This competition for binding effectively acts to reduce the concentration of $[S]_C$ available to bind to the pump and is dependent on the growth rate (Klumpp & Hwa, 2014; Scott *et al*, 2014; Hui *et al*, 2015). Although this would affect any absolute estimates of binding rates (resulting in underestimates of $k_1$), it does not affect comparisons across TetB variants since the host strain remains unchanged. Adding more parameters to our model, however, could result in an underdetermined system, leading to the inability to extract significant information. The current model recapitulates the most salient features of the system from the most readily accessible and scalable experimental assays.

The present work can be expanded to characterize other MFS efflux pumps whose substrate is bacteriostatic, allowing for the plot of steady bacterial growth decrease as a function of substrate concentration. Although detailed information regarding substrate diffusion is known for TET, this information should not be necessary for the determination of relative changes of physicochemical properties within a single substrate. One application for this model

would be the characterization of other MFS single or multidrug anti-porters to determine which residues are important for drug binding. For multidrug transporters, it would be interesting to see if the residues involved in binding remain the same for all transported drugs. An additional application is the screening for specific pheno-types to possibly aid in determining structures for other MFS transporters. One of the difficulties in determining the structure of MFS transporters is the potential conformational heterogeneity of the transporter once purified. Wright and Tate described their char-acterization of a *tet(B)* mutant library in search of a variant locked within a conformation state that has the ability to bind tetracycline but not efflux it, which may make crystallization and diffraction less challenging (Wright & Tate, 2015). Our characterization method can be used to screen through the mutants to reveal which variant(s) have this desired phenotype. Of our tested mutants, it is possible that *tet(B)$^{G366C}$* would be a good candidate for structural studies as it binds substrate similar to wild type but has reduced transport abilities.

Taken together, our work demonstrates that physicochemical protein properties can be determined from *in vivo* high-throughput fitness assays. The success of this model opens the possibility of characterizing protein libraries and challenging to study proteins in a rapid timeframe as protein overexpression and purification are not essential for characterization.

# Materials and Methods

## Construction of *tet(B)* expression construct

The *pBAD/tet(B)/rrnB/pET28b* (+) expression construct was created using restriction enzyme/ligation cloning within the pET28b (+) vector (Novagen). Both the *pBAD* promoter and *rrnB* terminator were amplified from the pTARA plasmid, a gift from the Kathleen Matthews laboratory at Rice University (Wycuff & Matthews, 2000). Tetracycline efflux pump *tet(B)* was amplified from a genomic prep from XL1 Blue Supercompetent cells from Stratagene (Agilent Tech-nologies, Santa Clara, CA, USA) as these cells naturally harbor the Tn10 transposon, which contains *tet(B)*. Primer pairs are listed in Dataset EV1. The genomic prep was performed using the MO BIO Ultraclean Microbial DNA Extraction Kit (MO BIO, Carlsbad, CA, USA). A ribosome binding site (RBS) sequence (GAGAGACTCCTC TCCATAACGAGGCCTATAAAC) was inserted upstream of *tet(B)*. The RBS sequence was designed with the Salis Lab RBS calculator and had an arbitrary strength of 20,000 (Salis *et al*, 2009). Stronger RBS sequences that induced much greater levels of TetB were found to also increase fitness costs, and therefore, we decided upon a more moderate strength for the RBS sequence.

## Construction of *tet(B)* chromosomal variants

TetB variants were created using site-directed mutagenesis performed on *pBAD/tet(B)/rrnB/pET28b* (+) template with Phusion High-Fidelity DNA Polymerase (New England Biolabs). The primer pairs used for mutagenesis are listed in Dataset EV1. Successful mutagenesis was confirmed by DNA sequencing. These constructs were then subcloned into pGRG36 between PacI and XhoI restriction sites. Insertion was confirmed using colony PCR and sequencing.

## Construction of chromosomal insertion strains

Integration of *tet(B)* and variants into the chromosome of *E. coli* strain BW25113 [F-, Δ(araD-araB)567, ΔlacZ4787(::rrnB-3), lambda-, rph-1, Δ(rhaD-rhaB)568, hsdR514] was performed using a transposition-based approach that uses Tn7 genes within pGRG36 to site-specifically insert the *pBAD/tet(B)/rrnB* construct into the Tn7 attachment site downstream of the highly conserved glmS gene following the method described by McKenzie & Craig (2006). First, *tet(B)$^{WT}$* and variant constructs were subcloned from pET28b(+) into pGRG36. pGRG36 is temperature sensitive and encodes an ampicillin resistance gene. Once successful transposi-tion is confirmed, the cells can be cured of pGRG36 by increasing the growth temperature from 30 to 42°C. Curing was also con-firmed by replica plating on agar plates containing ampicillin. Liquid cultures of transposed and cured strains were stored at −80°C in 15% glycerol.

## Determination of absolute growth rates for *tet(B)* chromosomal strains

Growth rates of *E. coli* BW25113 cells with and without a chromo-somal copy of *tet(B)$^{WT}$* or variant were performed in non-tissue-cultured 96-well plate with low evaporation lid (Corning Costar 3595). Glycerol stocks were streaked onto LB-glucose plates where glucose was used to repress the pBAD promoter and subsequently expression of *tet(B)* or variant. Single colonies were grown in 1 ml of 2% arabinose-supplemented LB in a 96-deep well plate for 24 h (±1 h) in an orbital shaker at 225 rpm at 37°C. Cultures were then diluted 1:100 in antibiotic + 2% arabinose-supple-mented LB media (10 μl culture + 990 μl media); 150 μl of this mixture then added to a 96-well plate where OD$_{600}$ was measured every 5 min for 20–24 h using the Epoch 2 Microplate Reader (BioTek Instruments).

## Membrane preparation to estimate protein concentration

Glycerol stocks of BW25113 cells containing a chromosomal copy of *tet(B)$^{WT}$* or variant were streaked onto glucose-supplemented LB plates and incubated at 37°C to obtain single colonies. Single colo-nies were then grown 2% arabinose-supplemented LB for 24 (±1 h) h at 37°C shaking at 225 rpm. Cells were diluted 1:100 in fresh 2% arabinose-supplemented LB (1.25 ml culture in 125 ml of LB). Growth was monitored by checking OD$_{600}$ every 30 min. Once the cells reached mid-exponential growth phase (approximately OD$_{600}$ 0.4–0.6), 60 ml of the culture was spun down at 4,000 *g* for 15 min. The cell pellet was then weighed and stored at −80°C. Bacterial membranes were prepared by freeze-thawing the cell pellet 3× in liquid nitrogen/37°C for 10 min followed by resuspension in lysis buffer to 25 mg/ml [50 mM Tris pH 7.5, 1 M NaCl, 0.3 mM DTT, 0.2 mM PMSF, 20% glycerol, 0.05% Tween 20, and 1× protease inhibitor (Roche)]. Sonication using a microtip was performed at 60% duty and six output twice for 30 s and then spun at 2,000 *g* for 15 min to remove cell debris. The supernatant was then spun at 18,000 *g* for 45 min to pellet the membrane fraction of the cell. The membrane fraction was then resuspended in lysis buffer to a concentration of 10 mg/ml and immediately used for Western blot analysis.

## Determination of protein concentration in membrane

Western blots on bacterial membrane fractions were used to estimate the amount of protein in *tet(B)* variant constructs relative to wild type; 45 μg of membrane samples was run on a 10% SDS–PAGE gel for 1 h. Protein samples were then transferred onto PVDF membrane for 2 h using a Bio-Rad semidry transfer system and blocked with 5% non-fat milk in TBST for 1 h. The membrane was then washed with TBST and incubated overnight with 1:5,000 dilution of an antibody for the last 15 amino acids on the cytoplasmic C-terminal tail of TetB (Yamaguchi *et al*, 1990). The TetB antibody was created by GenScript using the peptide sequence published by Yamaguchi *et al* (LTPQAQGSKQETSA) with the exception that a cysteine was added to the N-terminal of the peptide to aid in keyhole limpet hemocyanin (KLH) conjugation. The membrane was then washed and incubated with 1:5,000 dilution of anti-rabbit secondary antibody for 90 min (Pierce Antibodies). The membrane was again washed and imaged using SigmaFast BCIP/NBT tablets. ImageJ software was used for quantification.

## Creation and screening of plasmid *tet(B)* mutant library

The pBAD-driven *tet(B)* expression construct in pGRG36 was amplified and inserted into low-copy-number, ampicillin-resistant plasmid pSC101 using restriction ligation cloning in BW25113. Agilent Gene Morph II EZ Clone Domain Mutagenesis Kit (cat# 200552) was used to create a low rate, 1–4 mutations/kb, *tet(B)* mutant library using 640 ng of target DNA. The primers used for creation of the library amplified only *tet(B)* and are listed within Dataset EV1. The resulting library was transformed into BW25113 using electroporation and plated on 25 μg/ml TET. The following day, replica plating was performed on 250 and 300 μg/ml TET to screen for possible loss- and gain-of-function variants. On a single 96-well plate, 44 single colonies from the library were further screened against four WT single colonies by a growth assay at 100 μg/ml. Seven strains, which had a growth rate different than WT, were sequenced and further characterized by growth analysis at a wide range of TET concentrations. Growth rate setup of the plasmid strains were performed identical to that of the chromosomal strains with the exception that 100 μg/ml ampicillin was also added to liquid and agar media to ensure selection of the plasmid backbone pSC101.

## Agar MIC comparison of BW25113 containing chromosomal *tet(B)* versus plasmid *tet(B)*

Agar MIC testing was performed on BW25113 containing pSC101 expressed *tet(B)* to serve as a metric/guide for replica plating to be done on the pSC101 expressed *tet(B)* mutant library. The lowest concentration of TET that inhibited growth of WT was then used to for replica plating in an effort to discern between possible gain-of-function or loss-of-function TetB variants. First, glycerol stocks of BW25113 expressing *tet(B)* from the chromosome or pSC101 plasmid were streaked onto 0.18% glucose-supplemented LB agar to repress the pBAD arabinose-inducible promoter. Three single colonies for each strain were then grown for 24 h in 1 ml of 2% arabinose-supplemented LB in a deep well plate; 2 μl of culture was then spotted onto LB arabinose plates containing 0, 100, 150, 200, 250, or 300 μg/ml TET and incubated for 24 h at 37°C. The pSC101 expressing *tet(B)* strain grew at 250 μg/ml but not 300 μg/ml.

## Mathematical analysis

All computation and data analyses were done in Matlab R2013b using custom written code (Code EV1). $OD_{600}$ data from growth rate assays were used to compute growth rates for each drug concentration (Appendix Fig S11). A value of 0.086 was subtracted from all data points to zero measurements against the $OD_{600}$ of the media. The slope of the logarithmically transformed $OD_{600}$ data at each time point was approximated using a moving window. Then, we took the measured slope at the first inflection point of the data as our measured growth rate. This inflection point appears as a peak in the first derivative. The parameters used in the analysis where the window size for averaging and the bin size used in finding the peaks in the data. Changing the window size and bin size was used to remove stochasticity in growth and measurement noise. To avoid bias, the parameters were held constant across replicates. To fit the models, the data were then normalized by dividing all data points by the average growth rate at zero drug concentration across the replicates. Parameter fitting was done by minimizing the mean square error using the Matlab optimization function fmincon (Code EV2).

## Determination of error within global parameters *A* and *B*

Global parameters *A* and *B* are used in downstream analysis of cells expressing *tet(B)* leading to a propagation of errors. Figure 1B shows the fits generated by the model in response to TET, DOX, and MCN. These are the least squares (LS) fits that correspond to the mean of the distribution. We consider the range of possible values of *A* and *B* that would fit any combination of the measured growth rates at each concentration. The growth rate equation (1), $GR = 1 - \frac{[S]_C^B}{A^B + [S]_C^B}$, changes continuously in parameters *A* and *B*; therefore, we can exploit this fact to reason that the ranges of *A* and *B* can be found by fitting parameters to two sets of data consisting of the all the maximum and minimum measured growth rates (i.e., the upper and lower bounds of the data). Table 1 shows the calculated range of the parameters along with the median found from the range and the least squares fit for comparison. The median values are close to the least squares fit values revealing that our distribution is most likely symmetric. We approximate the distribution of *A* and *B* with a gamma distribution centered at the least squares fit value. We approximate the standard deviation from the calculated spread. We assume that the experimental data sets obtained are contained within one standard deviation of the mean. We assume that not enough measurements were taken to see data on the outskirts of the distribution. For our sample size, the predicted standard deviation is $\sigma \approx (A_{max} - A_{min})/2$. We use the results in this section for our analysis of TetB variants.

## Creation of TetB homology model

A theoretical model of the TetB tertiary structure was created using Phyre2 based upon its sequence homology with the *E. coli* MFS transporter YajR (PBD ID: 3wdo; Kelley *et al*, 2015). YajR was selected for the homology model due to its high sequence similarity to TetB (32%) with the resulting model displaying 100% confidence

according to Phyre2. Based on protein similarity alone, it is believed that YajR is also a drug:H$^+$ antiporter (Jiang *et al*, 2013). For the figures, ribbon diagrams of the model were created using the PyMOL Molecular Graphics System, version 1.8 Shrödinger, LLC.

**Expanded View** for this article is available online.

## Acknowledgements

The authors thank Dr. Milya Davlieva for guidance with the Western blots and Dr. Aaron M. Collier for help with creating PyMOL structures. We also thank Drs. John S. Olson and Jonathan J. Silberg for their advice. This work was supported by the National Institutes of Health Grant R01AI080714 (Y.S.), by the National Institutes of Health, through the joint NSF/NIGMS Mathematical Biology Program grant R01GM104974 (M.R.B.), the Robert A. Welch Foundation grant number C-1729 (M.R.B.), the California Alliance Postdoctoral Fellowship (M.M.G.), the University of California President's Postdoctoral Fellowship (M.M.G.), and a training fellowship from the Keck Center of the Gulf Coast Consortia Houston Area Molecular Biophysical Program under the NIGMS T32GM00828 (A.P.).

## Author contributions

YS conceived of the idea, and YS, MRB, and AMP conceived of the experiments. MMG produced mathematical models and analysis. AMP, PK, and EO'B-G contributed to experimental work. AMP, MMG, and YS contributed to the writing of the manuscript. MRB advised on mathematical derivations.

## Conflict of interest

The authors declare that they have no conflict of interest.

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
