## [Review Process File · Molecular Systems Biology]

Using cellular fitness to map the structure and function of a Major Facilitator Superfamily effluxer

Anisha M. Perez, Marcella M. Gomez, Prashant Kalvapalle, Erin O'Brien-Gilbert, Matthew R. Bennett, and Yousif Shamoo

Review timeline:

Submission date:	15 March 2017
Editorial Decision:	26 April 2017
Revision received:	25 July 2017
Editorial Decision:	22 August 2017
Revision received:	19 October 2017
Editorial Decision:	17 November 2017
Revision received:	29 November 2017
Accepted:	1 December 2017

Editor: Maria Polychronidou

Transaction Report:

1st Editorial Decision

26 April 2017

Thank you again for submitting your work to Molecular Systems Biology. We have now heard back from the three referees who agreed to evaluate your study. As you will see below, the reviewers appreciate that the presented approach seems interesting. However, they raise a series of concerns, which should be carefully addressed in a revision of the manuscript.

The reviewers' recommendations are rather clear so I think that there is no need to repeat the points listed below. Of course, feel free to contact me in case you would like to discuss any particular point in further detail.

REVIEWER REPORTS

Reviewer #1:

Just as biochemical properties of a protein may be used to infer cellular fitness, cellular fitness may be used to infer biochemical properties of a protein. Compared to more traditional biochemical analysis with purified protein, cellular fitness is easier to measure and captures protein activity in a living cell.

The authors demonstrated proof of principle for this method with TetB, an efflux pump protein that confers resistance to tetracycline antibiotics. In particular, they developed an efflux model able to

take fitness data and output biochemical parameters for substrate binding and pumping efficiency across a number of mutants. Using this model, the authors were able to associate growth rate with kinetic parameters of TetB and its variants to certain antibiotics. As a result, by fitting growth rate vs. antibiotics concentrations, the authors were able to infer lumped kinetic parameters associated with each variant of TetB. The lumped kinetic parameters are further disentangled by accounting for membrane protein abundance and by running simulations with randomized parameters. They further demonstrate that the method can be used to screen for mutated TetB by associating mutations with a change in kinetic parameters.

By and large, this is an exciting work and I would recommend its publication after clarifications and revisions in several aspects.

The conceptual framework:

The concept of deriving specific biochemical parameters from high-level measurement of cellular fitness is very desirable. While the work focuses on efflux proteins, the approach is generally applicable to other proteins (or processes that influence growth) of interest. In this regard, I feel the main message of the paper could have been made clearer. For instance, it can be argued that, once we accept the model (eqs 1-10; but see below for some technical caveats) as being structurally appropriate for the class of proteins, pump parameters (γ and κ) can be treated as the intrinsic properties of the pump protein in the particular host. If so, the significance of further dissecting these phenotypic parameters is not particularly clear. This issue is also tied to the clarity of the presentation of the modeling framework, as noted below.

Modeling framework:

1. The modeling assumptions overall make sense. One potential caveat is that it doesn't consider the binding of Sc to a target (ribosome). This additional binding kinetics would compete with the pump protein for free Sc, which could affect the modeling result. I don't think this will change the overall framework but may affect estimates of pump parameters (or interpretation of pump parameters). This issue should be clarified.

2. Eq. (1) is used to back calculate [S]c from growth rate for strain expressing TetB. However, parameters in Eq. (1) are calculated from a strain with no expression of TetB. The main assumption for this back calculation is that growth rate as a function of [S]c for no TetB is the same as that with full induction of TetB. However, there is no evidence shown in the manuscript that validates this key assumption.

3. I am confused about the diffusion terms in Eqs 4 and 5. Why adding a coefficient of 0.5 to [S]_P and [S]_C? This is related to the schematic in Figure 1A ([S]_P = 2[S]_M; [S]_C = 4[S]_M). In the Supplement, the authors cited the literature for the concentration ratios but I'm not sure how the concentration ratio is interpreted (in such a way to ensure conservation of mass during transport).

4. Another major assumption of this method is that the protein of interest should not have any unintended effects (toxicity, burden, etc.) to growth rate. Especially for TetB mutants, if the effluxers have unintended effects, the interpretation of the phenotypic parameters (γ and κ) should be different.

As noted above, accounting for these additional factors will likely change the definition of pump parameters, without affecting the overall framework. But these issues should be clarified in a revision.

Minor comments:

1. The authors should note a relevant paper on analyzing bacterial responses to antibiotics: Tan et al Mol Syst Biol 2012. The inoculum effect and band-pass bacterial response to periodic antibiotic treatment.

2. If I understood the math analysis correctly, [S]_M is the concentration of antibiotics used to treat the samples. Maybe it's better to make this point clearer.

3. What explains the intercepts of fitted lines in Figure 5A? According to Eq. (9), (10) and (12), no intercepts are expected.

4. How the Taylor expansion (Eq. 11) was derived is not apparent. It would be helpful to include the steps to acquire Eq. 11 in the supplement.

5. Figure 4, y axis should be ratio to WT, not percentage.
6. Why is the distribution used for A and B in Figure 5 Gamma distribution, instead of Gaussian?

Reviewer #2:

The idea of using a macroscopic observable (such as growth rate) as a proxy for inferring biochemical parameters of proteins is intriguing and could facilitate better understanding of protein functions and mapping from genotype to phenotype. In this particular study, Perez and coworkers focused on the determination of biochemical properties of the MFS tetracycline efflux pump (TetB) from the growth rate of bacteria exposed to various concentrations of antibiotic. Specifically, the authors aimed to determine two phenomenological parameters of the pump from the measurements of dose-response curves: substrate binding affinity and pumping efficiency. Basing on the mathematical model, which connects the global cellular parameters to the parameters of studied efflux pump, they have further characterized a set of mutated variants of TetB.

In general, the questions addressed in this study are certainly timely and relevant and the approach mostly appears sound (especially the strain construction and biochemical analysis). This topic should be of interest to a broad readership. However, I have several concerns regarding both the modeling approach and technical aspects of the experiments (see below) that need to be clarified to corroborate the proposed claims and increase confidence in the reliability of the data and the conclusions that are drawn.

Specific issues:

1. This work needs to be put more carefully in the context of previous studies; specifically, the cited reference by Deris et al, 2013 is of paramount importance to this kind of work. It also includes an important physiological relation between gene expression and growth rate, which is omitted in this manuscript. This reference also addresses a similar question: how is fitness affected by the molecular details of the resistance mechanism? However, it leaves the protein intact but varies its expression whereas this study perturbs the protein. These two approaches are likely equivalent -- in both studies, the model assumes that the efflux/degradation of the drug is proportional to the product of the protein abundance and the protein-specific rate. Hence, it does not matter if the abundance or the rate is varied. The Deris paper took into account bacterial growth laws that capture how the composition of the proteome changes as a result of antibiotics and growth rate changes. This is not done in the present work and it is not clear why: taking the growth laws into account is increasingly becoming standard in the field and seems more powerful than the approach taken by the authors. It would be crucial to clarify these points and explain any advantages of the present approach.
2. The dilution of 1:100 from the overnight pre-culture is quite low. Such dilution gives a very limited window of steady exponential growth, especially since the experiment is started from a stationary culture. It typically takes ~7 generations to reach a steady state of balanced exponential growth and a 100-fold dilution provides less than 7 generations till stationary phase is reached again! I would strongly suggest redoing the experiments for the BW25113 strain using e.g. a 1:2000 dilution and comparing the results to verify the findings.
3. I did not fully understand the determination of the absolute growth rate and why well-established methods were not used (fitting a line to logarithmically transformed OD values). This doubt in the method of the growth determination was further enhanced by looking at Figure S3 and Figure 7 where the absolute growth rate is reported as ~0.8/h. This value seems more than two-fold too low for fast growing strains (such as BW25113 used here) in rich LB medium at 37C. Too low growth rates can be a consequence of inappropriate OD background subtraction. The latter is not mentioned in the methods; was background correction carried out? If so, how was this correction performed? In Figure S5 where some exemplary growth curves are depicted (for batch culture conditions) the background OD value is around ~0.05. If correction was performed and the observed growth rate is as reported, I would be skeptical that a steady state of balanced exponential growth has been reached in these experiments (see previous point).
4. When reporting fits to the data, quantitative measurements of goodness of fit (e.g., R squared and

relevant statistics) should be provided. This is particularly important since the agreement of model and data is sometimes not great (see e.g. red line and data points in Fig. 7A); in these cases, it is not clear if interpreting differences in the parameters obtained from the fits is meaningful.

5. The assumption that the antibiotics enter the cell simply by diffusion across the membrane severely limits the applicability of the model. While this probably holds for TET and DOX used by the authors, the uptake of many other antibiotics is likely more complicated (e.g. the uptake of aminoglycosides depends on the proton motive force). This should be discussed. It also needs to be explained how diffusion across the membrane can lead to a 4-fold higher drug concentration inside the cell (e.g. text above eq. (1)); I can imagine explanations for this but as currently described, it is confusing since diffusion alone should equilibrate the intracellular and extracellular drug concentrations.

6. The determination of protein levels by Western blots may not be sensitive enough for the quantitative comparison to the model that is crucial here. It is generally difficult to detect changes in protein levels that are smaller than two-fold with this technique. These measurements were done in triplicate and the replicates show considerable variability; e.g. in Figure 4, one of the replicates of E192Q is lower than all replicates for V339C, yet these data are not interpreted as E192Q having a lower expression level than WT. A statistical analysis is needed here to validate that the relatively subtle changes in protein levels are significant.

7. The sentence "This interaction is represented through A which roughly corresponds to the apparent K_d of the antibiotic binding to the ribosome ...," just below Eq. (1) is over-interpreting the meaning of the given phenomenological equation. I would suggest avoiding such claims if no literature is cited to support it. It is also confusing that K_d and A are used interchangeably in the following sentences.

8. I think that Eq. (S20) should be provided in the main text rather than in the supplementary material; it provides an analytical expression for cytoplasmic antibiotic concentration and together with Eq. (1) fully specifies the model. It would further be good to use a consistent notation of quantities in the derivation of mathematical model.

Minor points:

1. I do not fully understand the meaning of number 20,000 in the following sentence from Material and Methods: "A ribosome binding site (RBS) sequence, (GAGAGACTCCTCTCCATAACGAGGCCTATAAAC) from the Salis Lab RBS calculator of 20,000 was added to tet(B) through additional PCR amplification (Salis et al., 2009)."
2. The references for plasmids pET28b(+) and pSC101 should be included.
3. Is the arabinose repressor araC encoded on the plasmid (for the plasmid library) or is the chromosomal copy sufficient for tight repression?

Reviewer #3:

Impact

In this manuscript, Perez et al describe a model that provides a quantitative link between organismal fitness and the physicochemical properties of an efflux pump that mediates antibiotic resistance, and explore this relationship to establish a platform for evaluating functional parameters of protein variants from growth measurements. As a proof of concept, novel mutations generated by error-prone PCR were characterized according to kinetic parameters that reflect pumping efficiency and drug affinity. This "reverse" approach is innovative and should provide a quite significant advance both towards a better understanding of the molecular determinants of antibiotic resistance evolution and in the study of membrane proteins, which are traditionally challenging to characterize in vitro. Nevertheless, it is not very clear how this work could offer a significant contribution to the field of molecular systems biology. The formalism presented here is similar to what was described in a

previous publication by same group (Walkiewicz, 2012), where the tetracycline concentration in cytoplasm was modeled taking into account the functional properties of a tetracycline-degrading enzyme (TetX2), instead of an efflux pump. In that aspect, the progress brought by this work is somewhat limited. Therefore I believe that this paper would have much more impact if published instead in journals dedicated to the study of protein biophysics/biochemistry and/or molecular evolution.

Overall quality

This paper is very solid. The experiments are well designed and the methodologies are described with sufficient detail. The results have been carefully analyzed and presented in a clear manner. I don't have any major concern, only a few minor suggestions.

- Growth rates are described as being calculated from the 1st derivative of untransformed OD data, but I believe the logarithm of OD should be used instead.

- The authors use an arabinose-induced promoter to drive the chromosomal expression of tet(B), and remove the regulatory elements to avoid complicated expression dynamics. Although this is understandable, it still might be useful to provide to readers a sense of how the presence of TetR changes the fitness landscape; e.g. by comparing the results shown in figure 1 with similar experiments performed using the endogenous regulatory machinery.

- Parameter A in equation 1 reflects the drug interaction with the ribosome and it appears to be independent of processes that influence intracellular drug concentration (e.g. efflux vs degradation). Can the parameter A obtained in this work be compared with the corresponding parameter measured in previous work (Walkiewicz et al 2012)?

- Can differences in protein abundance shown in figure 4 be interpreted taking into account the position and type of mutations and their perceived impact on folding stability of this integral membrane protein?

1st Revision - authors' response

25 July 2017

MSB -17-7635: Author responses to reviewers

Reviewer #1:

Just as biochemical properties of a protein may be used to infer cellular fitness, cellular fitness may be used to infer biochemical properties of a protein. Compared to more traditional biochemical analysis with purified protein, cellular fitness is easier to measure and captures protein activity in a living cell.

The authors demonstrated proof of principle for this method with TetB, an efflux pump protein that confers resistance to tetracycline antibiotics. In particular, they developed an efflux model able to take fitness data and output biochemical parameters for substrate binding and pumping efficiency across a number of mutants. Using this model, the authors were able to associate growth rate with kinetic parameters of TetB and its variants to certain antibiotics. As a result, by fitting growth rate vs. antibiotic concentrations, the authors were able to infer lumped kinetic parameters associated with each variant of TetB. The lumped kinetic parameters are further disentangled by accounting for membrane protein abundance and by running simulations with randomized parameters. They further demonstrate that the method can be used to screen for mutated TetB by associating mutations with a change in kinetic parameters.

By and large, this is an exciting work and I would recommend its publication after clarifications and revisions in several aspects.

The conceptual framework:

The concept of deriving specific biochemical parameters from high-level measurement of cellular fitness is very desirable. While the work focuses on efflux proteins, the approach is generally applicable to other proteins (or processes that influence growth) of interest. In this regard, I feel the main message of the paper could have been made clearer. For instance, it can be argued that, once we accept the model (eqs 1-10; but see below for some technical caveats) as being structurally appropriate for the class of proteins, pump parameters (γ and κ) can be treated as the

intrinsic properties of the pump protein in the particular host. If so, the significance of further dissecting these phenotypic parameters is not particularly clear. This issue is also tied to the clarity of the presentation of the modeling framework, as noted below.

Modeling framework:

1. The modeling assumptions overall make sense. One potential caveat is that it doesn't consider the binding of Sc to a target (ribosome). This additional binding kinetics would compete with the pump protein for free Sc, which could affect the modeling result. I don't think this will change the overall framework but may affect estimates of pump parameters (or interpretation of pump parameters). This issue should be clarified.

>> Thank you. We certainly agree with reviewer that the ribosome is a sink for tetracyclines. We considered the ribosome binding events but noted that the additional complexity of the model did not reward us with additional insights into pump performance. Including the steady state dynamics of ribosome binding would lead to a degradation term that is proportional to the number of ribosomes. However, the number of ribosomes changes with growth. Although there are several excellent papers relating ribosomes to growth rates we considered whether estimating or measuring ribosome pools would significantly alter our ability to compare effluxer performance. Since we are only considering relative changes in the pump parameters then we think it is reasonable to omit these dynamics since they do not change across the different mutants given that it's a characteristic of the host strain only. However, we agree that the point is a good one and we have added text to the Discussion section describing this consideration and appropriate references.

New text in Discussion paragraph 8:

“Additionally, we have omitted the binding of cytoplasmic drug [S]_C to the ribosomes. This competition for binding effectively acts to reduce the concentration of S_C available to bind to the pump and is dependent on the growth rate (Hui et al., 2015; Klumpp et al., 2014; Scott et al., 2014). Although, this would affect any absolute estimates of binding rates (resulting in underestimates of k₁), it does not affect comparisons across mutant TetB variants since the host strain remains unchanged.” <<

2. Eq. (1) is used to back calculate [S]_c from growth rate for strain expressing TetB. However, parameters in Eq. (1) are calculated from a strain with no expression of TetB. The main assumption for this back calculation is that growth rate as a function of [S]_c for no TetB is the same as that with full induction of TetB. However, there is no evidence shown in the manuscript that validates this key assumption.

>> Corrected: The growth rates are plotted as a function of [S]_M. Following Minor Comment 2 we have made this distinction clearer in the text.

The reviewer brings up an important point in our assumption showing that the fitness cost/ effects produced by expression of TetB is minimal in the background of our host strain BW25113. To calculate this, we used the average of the raw, non-normalized growth rates in zero drug concentration. We used the growth rates in the absence of drug from the TET, DOX and MCN growth assay plates- providing a total of 9 single colony replicates for the *tet(B)* strains and 12 single colony replicates for BW25113 host strain. To determine the fitness costs of TetB and the chromosomal variants within our host strain BW25113 we used the following equation:

$$\text{Percent Fitness Cost} = 1 - \left(\frac{GR^{TetB}}{GR^{BW25113}} \right) * 100$$

Insertion of a transgene within a bacterial system can often lead to a decrease in fitness of the cellular system. In our system, this cost was generally between 3% and 4%. The strain expressing *tet(B)^{L9F}* displayed a modestly larger fitness cost, approximately 4.7%, compared to the other variants. We are also aware that even modest fitness costs will lead to the fixation of new adaptive mutations and therefore we were careful to limit growth rate measurements to as small a time period as possible. We do not expect these modest fitness costs to alter the outcome of the model however we have added text to clarify this point and included this data in the Appendix Figure S3. <<

3. I am confused about the diffusion terms in Eqs 4 and 5. Why adding a coefficient of 0.5 to $[S]_P$ and $[S]_C$? This is related to the schematic in Figure 1A ($[S]_P = 2[S]_M$; $[S]_C = 4[S]_M$). In the Supplement, the authors cited the literature for the concentration ratios but I'm not sure how the concentration ratio is interpreted (in such a way to ensure conservation of mass during transport). >> We follow the 1995 work of Thanassi and coworkers to guide our understanding of tetracycline diffusion across the Gram-negative bacterial membranes. The concentration ratios during steady state are due in part to the Gibbs-Donnan effect across the outer membrane. Only the charged tetracycline- Mg^{1+} complex can diffuse across the outer membrane into the periplasm. The tetracycline molecules fail to distribute evenly across the two sides of the membrane because of the charge across the membrane. The positively charged complex on the outside of the membrane leads to a continuing diffusion of the tetracycline even after concentrations are equal. Ultimately, resulting in the concentration of tetracycline- Mg^{1+} charged complex to accumulate to 2x within the periplasm (Thanassi *et al.*, 1995).

Tetracycline crosses the inner membrane as an uncharged species. An increase in pH from the periplasm to the cytoplasm causes the uncharged tetracycline to once again be concentrated 2x within the cytoplasm. Overall, tetracycline accumulates 4x in the cytoplasm compared to the media/extracellular space.

We have included text in Results section- A physiological model describing TetB efflux pump kinetics can be used to determine physicochemical parameters from fitness- paragraph 3.

It is likely that changes in the molecular structures of DOX and MCN will have potentially different pKs and therefore we are careful to not compare across antibiotics but only within a single antibiotic for the estimated parameters. We again note that since our work focuses on determining relative changes in pump parameters, the diffusion patterns detailed within the model are applied to all variants. Thus, the relative changes in pump parameters will not be impacted. For bacteriostatic antibiotics, which enter the bacteria through diffusion, our model and experimental framework allows for relative pump parameters of protein variants to be determined without prior knowledge of specific diffusion patterns which we feel is a strength of our system. <<

4. Another major assumption of this method is that the protein of interest should not have any unintended effects (toxicity, burden, etc.) to growth rate. Especially for TetB mutants, if the effluxers have unintended effects, the interpretation of the phenotypic parameters (γ and κ) should be different.

>> Please see the response to Comment 2 above. <<

As noted above, accounting for these additional factors will likely change the definition of pump parameters, without affecting the overall framework. But these issues should be clarified in a revision.

Minor comments:

1. The authors should note a relevant paper on analyzing bacterial responses to antibiotics: Tan et al Mol Syst Biol 2012. The inoculum effect and band-pass bacterial response to periodic antibiotic treatment.

>> Thank you for bringing this interesting paper to our attention. We have added it to our references. <<

2. If I understood the math analysis correctly, $[S]_M$ is the concentration of antibiotics used to treat the samples. Maybe it's better to make this point clearer.

>> We agree that this could be clearer and have added the text after the ordinary differential equations are presented.

“The input into the model is the antibiotic concentration in the media ($[S]_M$). This is the antibiotic concentration used in the growth rate assays. The ordinary differential equations then model changes in the antibiotic concentration in the periplasm ($[S]_P$) and cytoplasm ($[S]_C$) based on $[S]_M$ as well as pump kinetics”

We have also modified Figure Legend 1B for clarity as below:

OLD Figure 1B legend: “Normalized fitness functions of *E. coli* BW25113 in the absence of *tet(B)* and their corresponding model fits in the presence of TET, DOX, and MCN. Normalized triplicate data is shown for each antibiotic.”

NEW Figure 1B legend: “Normalized fitness functions of *E. coli* BW25113 in the absence of *tet(B)* and their corresponding model fits in the presence of varying concentrations of antibiotics TET, DOX, and MCN in the media ($[S]_M$). Normalized triplicate data is shown for each antibiotic.” <<

3. What explains the intercepts of fitted lines in Figure 5A? According to Eq. (9), (10) and (12), no intercepts are expected.

>> Before making an approximation, we can see from the Taylor expansion (formerly Eq. (11) now Eq. (24)), that there are additional terms that result in κ and γ having nonlinear relationship.

However, we show through Figure 5A that the system for most mutants is in a linear regime. As we perturb the measured values of global parameters A and B we find that κ and γ maintain a linear correlation. By looking at the equations we can see that the relationship will fail to be linear when either r , pump out rate, or k_1 , initial binding rate, are on the same order of magnitude as k_{-1} , unbinding rate. <<

4. How the Taylor expansion (Eq. 11) was derived is not apparent. It would be helpful to include the steps to acquire Eq. 11 in the supplement.

>> Thank you. All equations are now included within the main text and the Taylor expansion, Eq 11, is now Eq. 24. The steps to acquire this equation are detailed in Eq. 22 and Eq. 23. <<

5. Figure 4, y axis should be ratio to WT, not percentage.

>> Thank you, this has been fixed. <<

6. Why is the distribution used for A and B in Figure 5 Gamma distribution, instead of Gaussian?

>> Gaussian distribution allows for negative values and we want to ensure only positive definite values of A and B are considered. <<

Reviewer #2:

The idea of using a macroscopic observable (such as growth rate) as a proxy for inferring biochemical parameters of proteins is intriguing and could facilitate better understanding of protein functions and mapping from genotype to phenotype. In this particular study, Perez and coworkers focused on the determination of biochemical properties of the MFS tetracycline efflux pump (TetB) from the growth rate of bacteria exposed to various concentrations of antibiotic. Specifically, the authors aimed to determine two phenomenological parameters of the pump from the measurements of dose-response curves: substrate binding affinity and pumping efficiency. Basing on the

mathematical model, which connects the global cellular parameters to the parameters of studied efflux pump, they have further characterized a set of mutated variants of TetB.

In general, the questions addressed in this study are certainly timely and relevant and the approach mostly appears sound (especially the strain construction and biochemical analysis). This topic should be of interest to a broad readership. However, I have several concerns regarding both the modeling approach and technical aspects of the experiments (see below) that need to be clarified to corroborate the proposed claims and increase confidence in the reliability of the data and the conclusions that are drawn.

Specific issues:

1. This work needs to be put more carefully in the context of previous studies; specifically, the cited reference by Deris et al, 2013 is of paramount importance to this kind of work. It also includes an important physiological relation between gene expression and growth rate, which is omitted in this manuscript.

>> This is a good observation and we were certainly aware of this when we designed the system. It is for this reason that TetB is not under a constitutive promoter but rather, constant induction by arabinose which is not metabolized by our host cell line BW25113. Therefore, expression of TetB is not positively correlated with growth rate (i.e. expression of TetB does not go down with decreased growth rate). This was experimentally confirmed by Western Blot analysis of cell samples obtained at two different normalized growth rates (~0.6 and 0.25) in DOX which showed similar TetB expression pattern, following our protocol outlined in Materials and Methods, the cell samples were harvested at mid-exponential growth phase. <<

This reference also addresses a similar question: how is fitness affected by the molecular details of the resistance mechanism? However, it leaves the protein intact but varies its expression whereas this study perturbs the protein.

>> In response to Reviewer 1 Question 2, we show that the fitness effects of chromosomal TetB and variants are modest within our system. <<

These two approaches are likely equivalent -- in both studies, the model assumes that the efflux/degradation of the drug is proportional to the product of the protein abundance and the protein-specific rate. Hence, it does not matter if the abundance or the rate is varied. The Deris paper took into account bacterial growth laws that capture how the composition of the proteome changes as a result of antibiotics and growth rate changes. This is not done in the present work and it is not clear why: taking the growth laws into account is increasingly becoming standard in the field and seems more powerful than the approach taken by the authors. It would be crucial to clarify these points and explain any advantages of the present approach.

>> In the Deris paper, the efflux pump was constitutively expressed and so it was critical that the growth rate laws be considered. In our system, expression is induced and so does not decrease with growth rate. Of course, at really low growth rates we can expect that even induced gene expression may subside but we are most concerned with the range of antibiotics before growth rate drops dramatically as things are more complicated thereafter. It is in this drop off where Deris et al. 2013 see bistability and Tan et al. 2012 observe what they call the inoculum effect. <<

2. The dilution of 1:100 from the overnight pre-culture is quite low. Such dilution gives a very limited window of steady exponential growth, especially since the experiment is started from a stationary culture. It typically takes ~7 generations to reach a steady state of balanced exponential growth and a 100-fold dilution provides less than 7 generations till stationary phase is reached again! I would strongly suggest redoing the experiments for the BW25113 strain using e.g. a 1:2000 dilution and comparing the results to verify the findings.

>> Thank you for suggesting this experiment. The measured growth rates of the BW25113 strain in 2% arabinose supplemented media the absence of antibiotic at the 1:100 and 1:2000 dilution were nearly identical. Values are represented as the average growth rate of 4 single colony replicates +/- standard deviation- 1:100 dilution (0.0136 ± 0.00012); 1:2000 dilution (0.0137 ± 0.00009). As seen from the overlaid growth curves, the 1:2000 dilution growth curve has an extended lag phase when compared to the 1:100 growth curve but retains the same growth rate.

3. I did not fully understand the determination of the absolute growth rate and why well-established methods were not used (fitting a line to logarithmically transformed OD values). This doubt in the method of the growth determination was further enhanced by looking at Figure S3 and Figure 7 where the absolute growth rate is reported as $\sim 0.8/h$. This value seems more than two-fold too low for fast growing strains (such as BW25113 used here) in rich LB medium at 37C. Too low growth rates can be a consequence of inappropriate OD background subtraction. The latter is not mentioned in the methods; was background correction carried out? If so, how was this correction performed? In Figure S5 where some exemplary growth curves are depicted (for batch culture conditions) the background OD value is around ~ 0.05 . If correction was performed and the observed growth rate is as reported, I would be skeptical that a steady state of balanced exponential growth has been reached in these experiments (see previous point).

>> We apologize for the lack of clarity. Well-established methods were used and a line was fit to logarithmically transformed OD values. We suspect the slow growth rate is due to the addition of arabinose. Furthermore, the control experiment done in response to the previous comment shows that the dilution rate does not change the calculated exponential growth rate. Subtraction of background OD was not carried out nor is it needed. The OD background only determines the y-intercept of the line that is fit to the data but does not affect the calculated slope since growth rate is determined by relative change in OD over time.

New text Materials and Methods- Mathematical Analysis:

“The slope of the logarithmically transformed OD_{600} data at each time point was approximated using a moving window. Then we took the measured sloped at the first inflection point of the data as our measured growth rate.” <<

4. When reporting fits to the data, quantitative measurements of goodness of fit (e.g., R squared and relevant statistics) should be provided. This is particularly important since the agreement of model and data is sometimes not great (see e.g. red line and data points in Fig. 7A); in these cases, it is not clear if interpreting differences in the parameters obtained from the fits is meaningful.

>> Thank you for pointing this out and we agree that error should be indicated for our model fits. We have added Appendix Table S1 and Table S3 to include the error for the chromosomal and

plasmid variants, respectively. The error was calculated by taking the square root of the sum of the residual errors, squared. <<

5. The assumption that the antibiotics enter the cell simply by diffusion across the membrane severely limits the applicability of the model. While this probably holds for TET and DOX used by the authors, the uptake of many other antibiotics is likely more complicated (e.g. the uptake of aminoglycosides depends on the proton motive force). This should be discussed. It also needs to be explained how diffusion across the membrane can lead to a 4-fold higher drug concentration inside the cell (e.g. text above eq. (1)); I can imagine explanations for this but as currently described, it is confusing since diffusion alone should equilibrate the intracellular and extracellular drug concentrations.

>> Please see our response to Review 1 Question 3 where we go into further detail about tetracycline diffusion in relation to concerns about higher drug concentrations inside the cell. In the Discussion section paragraph 9 of the manuscript, however, we mention that prior knowledge on the molecules diffusion pattern need not be known to use this method as we determine relative changes of the variants to wild-type. Additionally, any other uptake mechanism would be captured in the growth rate characterization without TetB. For example, if it is known that a molecule's diffusion pattern varies depending on its concentration, we can still isolate the fold changes in pump kinetics. In regards to aminoglycosides, we failed to mention within the manuscript that the type of analysis proposed in the present manuscript can be performed when using bacteriostatic antibiotics or other small molecules that lead to a slow down of bacterial growth instead of leading to cellular death. In response, we have added the following text to the Discussion section paragraph 9:

“The present work can be expanded to characterize other MFS efflux pumps whose substrate is bacteriostatic, allowing for the plot of steady bacterial growth decrease as a function of substrate concentration.” <<

6. The determination of protein levels by Western blots may not be sensitive enough for the quantitative comparison to the model that is crucial here. It is generally difficult to detect changes in protein levels that are smaller than two-fold with this technique. These measurements were done in triplicate and the replicates show considerable variability; e.g. in Figure 4, one of the replicates of E192Q is lower than all replicates for V339C, yet these data are not interpreted as E192Q having a lower expression level than WT. A statistical analysis is needed here to validate that the relatively subtle changes in protein levels are significant.

>> We agree that quantitative Westerns blots are difficult to have confidence in below ~2-fold changes. We incorporated the uncertainty in Western blot measurements in the final analysis to make this clear. It is the variability in Western blots that hinders most our ability to make predictions as can be seen by some of the large error bars in Figure 6A. This may have not been clear since the derivation for the uncertainty propagation in the analysis is in the supplement in equations (S21)-(S26). These have been moved into the main text. We would also like to point out that because of the variability in measurements of protein levels in E192Q, Figure S8 shows that we cannot draw any statistically significant conclusions from this particular mutant. <<

7. The sentence "This interaction is represented through A which roughly corresponds to the apparent K_d of the antibiotic binding to the ribosome ...," just below Eq. (1) is over-interpreting the meaning of the given phenomenological equation. I would suggest avoiding such claims if no literature is cited to support it. It is also confusing that K_d and A are used interchangeably in the following sentences.

>> Mathematically, the A parameter value is the cytoplasmic drug concentration, $[S]_C$ at which the normalized growth rate is at half maximal (0.5) and we adjusted the text accordingly and have removed ' K_d ' in a subsequent sentence. We also edited the text within Table 1 description to include the mathematical definition of parameter A .

We have modified the text in the Results section entitled “Baseline response of a cellular system without *tet(B)* is used to determine global parameters” to:

“This interaction is represented through A which corresponds to the cytoplasmic drug concentration, $[S]_C$, at which the normalized growth rate is half maximal. Additionally, global parameter A can roughly be interpreted as the apparent K_d of the antibiotic binding to the ribosome although it does not take into account other cellular sinks of the antibiotic (Walkiewicz et al, 2012).”

We removed the reference to K_d in the main text as noted below:

“It should be noted that our host strain *E. coli* BW25113, in contrast to many *E. coli* strains, has a higher intrinsic resistance to DOX and MCN than TET. This intrinsic resistance, however, is captured by our model and reflected in parameter A where this value is largest for MCN and smallest of TET (Table 1).”

Additionally, we have edited the Table 1 figure legend to reflect the updated main text:

Table 1. Global parameters A and B which describe the baseline response of the cellular system were computed from fitness functions of *E. coli* BW25113 in TET, DOX and MCN using Equation (1). “Parameter A has units of $\mu\text{g/mL}$ and corresponds to the cytoplasmic drug concentration, $[S]_C$, at which the normalized growth rate is half maximal. Parameter A can roughly be interpreted as the apparent K_d of the drug to the ribosome in addition to all other intracellular drug interactions.” <<

8. I think that Eq. (S20) should be provided in the main text rather than in the supplementary material; it provides an analytical expression for cytoplasmic antibiotic concentration and together with Eq. (1) fully specifies the model. It would further be good to use a consistent notation of quantities in the derivation of mathematical model.

>> Thank you, the editor has also suggested we move the equations within the supplement to the main text. All equations have been moved into the main text. <<

Minor points:

1. I do not fully understand the meaning of number 20,000 in the following sentence from Material and Methods: "A ribosome binding site (RBS) sequence, (GAGAGACTCCTCTCCATAACGAGGCCTATAAAC) from the Salis Lab RBS calculator of 20,000 was added to *tet(B)* through additional PCR amplification (Salis et al., 2009)."

>> We designed and tested ribosome binding sites of varying strengths using the Salis Lab RBS calculator. 20,000 refers to the arbitrary strength of the RBS from that calculator. We have changed the text to make this clearer. We have also now included the fitness effects to address any concerns about fitness costs associated with expression (Appendix Figure S3).

Revised Text in Materials and Methods- Construction of *tet(B)* expression construct:

"A ribosome binding site (RBS) sequence, (GAGAGACTCCTCTCCATAACGAGGCCTATAAAC) was inserted upstream of *tet(B)*. The RBS sequence was designed with the Salis Lab RBS calculator and had an arbitrary strength of 20,000 (Salis et al., 2009). Stronger RBS sequences that induced much greater levels of TetB were found to also increase fitness costs and therefore we decided upon a more moderate strength for the RBS sequence." <<

2. The references for plasmids pET28b(+) and pSC101 should be included.
Thank you, the sources for these plasmids have now been included.

3. Is the arabinose repressor *araC* encoded on the plasmid (for the plasmid library) or is the chromosomal copy sufficient for tight repression?

>> Within our cellular system, *araC* is encoded on the chromosome and is sufficient because our fitness experiments are conducted in saturating arabinose concentration (2%). This saturating arabinose concentration allows us to have a homogenous induced cellular population of TetB. <<

Reviewer #3:

Impact

In this manuscript, Perez et al describe a model that provides a quantitative link between organismal fitness and the physicochemical properties of an efflux pump that mediates antibiotic resistance, and explore this relationship to establish a platform for evaluating functional parameters of protein variants from growth measurements. As a proof of concept, novel mutations generated by error-prone PCR were characterized according to kinetic parameters that reflect pumping efficiency and drug affinity. This "reverse" approach is innovative and should provide a quite significant advance both towards a better understanding of the molecular determinants of antibiotic resistance evolution and in the study of membrane proteins, which are traditionally challenging to characterize in vitro.

Nevertheless, it is not very clear how this work could offer a significant contribution to the field of molecular systems biology. The formalism presented here is similar to what was described in a previous publication by same group (Walkiewicz, 2012), where the tetracycline concentration in cytoplasm was modeled taking into account the functional properties of a tetracycline-degrading enzyme (TetX2), instead of an efflux pump. In that aspect, the progress brought by this work is somewhat limited. Therefore I believe that this paper would have much more impact if published instead in journals dedicated to the study of protein biophysics/biochemistry and/or molecular evolution.

Overall quality

This paper is very solid. The experiments are well designed and the methodologies are described with sufficient detail. The results have been carefully analyzed and presented in a clear manner. I don't have any major concern, only a few minor suggestions.

- Growth rates are described as being calculated from the 1st derivative of untransformed OD data, but I believe the logarithm of OD should be used instead.

>> Thank you for pointing this out. Reviewer #2 also raised this concern. We do in fact use the logarithm of the OD but failed to describe it clearly. We have clarified this and added the text. Please see response to reviewer 2, specific issue 3. <<

- The authors use an arabinose-induced promoter to drive the chromosomal expression of tet(B), and remove the regulatory elements to avoid complicated expression dynamics. Although this is understandable, it still might be useful to provide to readers a sense of how the presence of TetR changes the fitness landscape; e.g. by comparing the results shown in figure 1 with similar experiments performed using the endogenous regulatory machinery.

>> This is a fascinating topic in its own right and would require significantly different experiments and modeling. In the present work, we explore how cellular fitness provides details which relate to a protein's physicochemical properties. It would seem too speculative for us to suggest how TetR associated dynamics could be incorporated within our model without actually doing the math to provide a testable basis for the speculation. Incorporation of TetR within our cellular system would not only make the mathematical model more complicated but it would bring us further from the hypothesis we sought to explore- can we use phenotype to estimate the biochemical properties of a single, challenging to characterize protein.

In a TetR/TetB system, the level of *tet(B)* repression by TetR is dynamic and would change depending on the concentration of tetracycline- where more tetracycline present would ultimately lead to more TetB within the bacterial system. The assumption made in our current model would no longer be valid- as TetB concentration would now be dynamic. To measure these dynamics, western blots at each of the 12 concentrations of drug used in our experiments would have to be performed- for all three drugs (TET, DOX, MCN). Fundamentally however it seems that such a study would be a body of publishable work in its own right. <<

- Parameter A in equation 1 reflects the drug interaction with the ribosome and it appears to be independent of processes that influence intracellular drug concentration (e.g. efflux vs degradation). Can the parameter A obtained in this work be compared with the corresponding parameter measured in previous work (Walkiewicz et al 2012)?

>> The A parameter from the current work vs the A parameter in our previous work should not be compared as they depend on (among other things) the media conditions i.e. arabinose (current work) vs no arabinose (Walkiewicz et al, 2012). The growth rate function is more akin to a lumped parameter mapping growth rate to intracellular antibiotic concentrations. We would recommend fitting A for any change in strain or environment. <<

- Can differences in protein abundance shown in figure 4 be interpreted taking into account the position and type of mutations and their perceived impact on folding stability of this integral membrane protein?

>> The short answer is no. In our previous studies on adaptive thermostability (Couñago et al., 2006; Peña et al., 2010) despite having high resolution crystal structures for the wild type protein and all the adaptive proteins, only actual measurements of stability by CD and differential scanning calorimetry proved useful.

Couñago, R., Chen, S., and Shamoo, Y. (2006). In Vivo Molecular Evolution Reveals Biophysical Origins of Organismal Fitness. *Molecular Cell* 22, 441–449.

Peña, M.I., Davlieva, M., Bennett, M.R., Olson, J.S., and Shamoo, Y. (2010). Evolutionary fates within a microbial population highlight an essential role for protein folding during natural selection. *Molecular Systems Biology* 6. <<

2nd Editorial Decision

22 August 2017

Thank you again for sending us your revised manuscript. We have now heard back from the two referees who were asked to evaluate your study. As you will see below, the reviewers think that the revisions performed have improved the study. However, they list a few remaining concerns, which we would ask you to address in a minor revision.

We have implemented a "model curation service" for papers that contain mathematical models. This is done together with Prof. Jacky Snoep and the FAIRDOM team. In brief, the aim is to enhance reproducibility and add value to papers containing mathematical models. Prof. Snoep's summary (*Model Curation Report*) is pasted below. As you already know from your email exchange with him, and will also see in the report below, there are some very minor issues, which we would ask you to fix when you submit your revision.

 REVIEWER REPORTS

Reviewer #1:

I'm overall satisfied with the responses and revisions.

However, I feel the presentation of model derivation can be improved for clarity and rigor. For example, derivation from Eq 22 to Eq 25 assumes that $k_{-1} \ll k_1$. The more appropriate assumption (in a way consistent with their derivation) appears to be $k_{-1} \ll r$ (so it would be accurate only to keep the 0-order term in the Taylor expansion). If so, one actually doesn't need Taylor expansion to jump from 22 to 25. Also, it appears that a coefficient of "2" should be included in the RHS of Eqs 22-24.

I trust the authors will make these final-stage and minor revisions/corrections.

Reviewer #2:

In the revised version, the authors have carried out additional experiments and analysis and have improved the text. However, there are some conceptual and technical issues remaining, which need further clarification.

1. Independent of how expression from a promoter is regulated by transcription factors, a large determinant of the expression level is the state of the cell as a whole. Specifically, even when the pBAD promoter is de-repressed by the addition of arabinose, its expression depends on the levels of cAMP. The latter exhibits a strong correlation with the growth rate - see, e.g., You et al, *Nature*, 500 (2013) doi:10.1038/nature12446. Thus, in the simplest case (ignoring the cAMP effect) when expression is fully de-repressed by saturating concentrations of inducer, the same effects arising from the physiological state of the cell affect expression as for the case of a constitutive promoter. Hence, I disagree that using the inducible promoter removes the need to take the growth rate-dependent effects on the gene expression into account (as in Deris et al, 2013 *Science*). These effects could be omitted if the negative autoregulation of the repressor were introduced (Klumpp et al, *Cell*, 2009; Scott et al. 2010, *Science*) but this would require redoing all the experiments. It would be important to revisit the mathematical model to address this point.

2. It is very good that the authors checked the effect of different dilution factors at inoculation and elaborated on the fitting procedure. However, the proposed procedure for obtaining the absolute growth rates seems incorrect. In contrast to the classical cuvette based absorbance measurements,

where background subtraction is done before each measurement by measuring the absorbance of a 'blank' cuvette (with growth medium only), plate reader data require determination of the background in a well-specific manner, e.g. by taking the median/mean of the first few measurements when the culture does not contribute to the density. The claim that background subtraction is not needed because it does not affect the growth rate (see response letter) is clearly wrong. For example, let the background (absorbance arising from the media itself, etc) be $bck=0.5$ and the first two measurements read $a1=0.501$ and $a2=0.51$ which were taken $dT=1$ hour apart.

Your procedure gives:

$$\ln(0.501) = -0.691$$

$$\ln(0.51) = -0.673$$

which in turn gives the growth rate $g = [\ln(a2) - \ln(a1)]/dT = 0.018/h$ with the doubling time of 2310 minutes.

Repeating the procedure with the background subtraction:

$$\ln(0.501 - 0.5) = \ln(0.01) = -4.61$$

$$\ln(0.51 - 0.5) = \ln(0.1) = -2.30$$

$$g = [\ln(a2 - bck) - \ln(a1 - bck)]/dT = 2.31/h$$
 with the doubling time of 18 minutes.

This is a consequence of the fact that the part of absorbance corresponding to the increasing number of bacteria is initially small and is virtually lost during the logarithmic transformation:

$\ln(bck + b \cdot \text{Exp}(gt)) - \ln(bck)$ if the $b \cdot \text{exp}(gt)$ is small which it is in the beginning. I hope that this makes the issue clearer; it can be easily resolved and this should certainly be done.

***Model Curation Report*:**

Technical curation for the mathematical models in MSB-17-7635R

The equation used for Figure 1 and 2 is well described in the manuscript, and I could readily reproduce the results shown in these figures. It was not clear from the manuscript how the growth rate shown in Figure 3 was calculated but after contacting the authors, and receiving their feedback, I could reproduce the simulation plots in Figure 3.

I would advice to indicate in the manuscript how the growth rate as plotted in Figure 3 was calculated, i.e. give an equation or state explicitly how it can be derived from Eq. 1. In addition the authors should refer to the table in which the non inhibited growth rates for the different mutant strains are listed.

2nd Revision - authors' response

19 October 2017

Point by Point Reviewer Response:

Reviewer #1:

I'm overall satisfied with the responses and revisions.

However, I feel the presentation of model derivation can be improved for clarity and rigor. For example, derivation from Eq 22 to Eq 25 assumes that $k_{-1} \ll k_1$. The more appropriate assumption (in a way consistent with their derivation) appears to be $k_{-1} \ll r$ (so it would be accurate only to keep the 0-order term in the Taylor expansion). If so, one actually doesn't need Taylor expansion to jump from 22 to 25. Also, it appears that a coefficient of "2" should be included in the RHS of Eqs 22-24.

I trust the authors will make these final-stage and minor revisions/corrections.

We thank reviewer 1 for their comments and have incorporated this more appropriate assumption.

Previous: If in fact the approximation $\varepsilon = k_{-1}/k_1 \ll 1$ holds, then we should expect that

$$\kappa = \frac{(r + k_{-1})}{k_1} \text{ correlates linearly with } \gamma = \frac{2rP_{tot}}{D_2}.$$

Revised: Substituting back in the definition for ϵ , we find that if $k_1 \ll r$, then a zero order approximation holds and we should expect that $\kappa = \frac{(r+k_1)}{k_1}$ correlates linearly with $\gamma = \frac{2rP_{tot}}{D_2}$.

Reviewer #2:

In the revised version, the authors have carried out additional experiments and analysis and have improved the text. However, there are some conceptual and technical issues remaining, which need further clarification.

1. Independent of how expression from a promoter is regulated by transcription factors, a large determinant of the expression level is the state of the cell as a whole. Specifically, even when the pBAD promoter is de-repressed by the addition of arabinose, its expression depends on the levels of cAMP. The latter exhibits a strong correlation with the growth rate - see, e.g., You et al, Nature, 500 (2013) doi:10.1038/nature12446. Thus, in the simplest case (ignoring the cAMP effect) when expression is fully de-repressed by saturating concentrations of inducer, the same effects arising from the physiological state of the cell affect expression as for the case of a constitutive promoter. Hence, I disagree that using the inducible promoter removes the need to take the growth rate-dependent effects on the gene expression into account (as in Deris et al, 2013 Science). These effects could be omitted if the negative autoregulation of the repressor were introduced (Klumpp et al, Cell, 2009; Scott et al. 2010, Science) but this would require redoing all the experiments. It would be important to revisit the mathematical model to address this point.

We agree with the reviewer that cAMP levels do play a role in pBAD promoter expression and that the state of the cell impacts expression level. Prior to performing the growth assays the cells are pre-treated in saturating inducer (2% arabinose) without any antibiotic for 24 hours allowing for an accumulation of TetB protein levels. Without pre-treatment, the growth rates displayed higher variability. We show via western blot below that TetB protein levels are similar for two different normalized growth rates (High ~0.6 and Low ~0.25) of TetB wild-type in BW25113 after 24 hour saturating arabinose treatment. Data was also shown to be similar for another tested, but discarded TetB construct of 35,000 RBS strength.

2. It is very good that the authors checked the effect of different dilution factors at inoculation and elaborated on the fitting procedure. However, the proposed procedure for obtaining the absolute growth rates seems incorrect. In contrast to the classical cuvette based absorbance measurements, where background subtraction is done before each measurement by measuring the absorbance of a 'blank' cuvette (with growth medium only), plate reader data require determination of the background in a well-specific manner, e.g. by taking the median/mean of the first few measurements when the culture does not contribute to the density. The claim that background subtraction is not needed because it does not affect the growth rate (see response letter) is clearly wrong. For example, let the background (absorbance arising from the media itself, etc) be $b_{ck}=0.5$ and the first two measurements read $a_1=0.501$ and $a_2=0.51$ which were taken $dT=1$ hour apart.

Your procedure gives:

$$\ln(0.501) = -0.691$$

$$\ln(0.51) = -0.673$$

which in turn gives the growth rate $g = [\ln(a_2) - \ln(a_1)] / dT = 0.018/h$ with the doubling time of 2310 minutes.

Repeating the procedure with the background subtraction:

$$\ln(0.501-0.5)=\ln(.01)=-4.61$$

$$\ln(0.51-0.5)=\ln(.1)=-2.30$$

$$g=[\ln(a_2-b_{ck})-\ln(a_1-b_{ck})]/dT=2.31/h \text{ with the doubling time of 18 minutes.}$$

This is a consequence of the fact that the part of absorbance corresponding to the increasing number of bacteria is initially small and is virtually lost during the logarithmic transformation:

$\ln(b_{ck}+b*\text{Exp}(gt))-\ln(b_{ck})$ if the $b*\text{exp}(gt)$ is small which it is in the beginning. I hope that this makes the issue clearer; it can be easily resolved and this should certainly be done.

We appreciate Reviewer 2's comment regarding subtracting the background within our growth rate assay OD600 measurements. We have now incorporated this change within our data by subtracting a value of .086 (measured OD600 reading of LB media used for the growth rate assays) to zero the data against the media. We find some changes in our findings but not in the general findings themselves. The correction required us to recalculate and plot growth rates and the figures have been updated accordingly. The changes performed in the main text are outlined below.

In section "Fitness functions of *tet(B)* variants can be modeled accurately to reveal relevant physicochemical properties of efflux pumps":

- The data no longer shows any evidence of bimodal response- we no longer see higher variability in growth rates at drop-off point. This was a very helpful improvement as we had been somewhat puzzled by the bimodal response. We were thus able to delete the passage below:

"For some variants, we observed large variability in growth rate at a critical transition point where fitness begins to drop quickly as indicated by a larger spread of the data (Appendix Fig S2). This is where cellular response to the presence of antibiotics is less understood and the reason is unclear. One potential explanation is a bimodal response experienced by bacteria in response to drug as demonstrated by Deris and coworkers (Deris et al, 2013). The drop-off point is also where Tan and coworkers find varying responses across populations based on initial cellular concentrations (Tan et al, 2012)."

-Growth rates of cells expressing *tet(B)^{E192A}* is now comparable to WT, therefore, Appendix FigS4 has been removed since it is no longer necessary and the sentence from the main text has been removed.

"Strains expressing *tet(B)^{E192A}* have a small but consistent decrease in fitness at intermediate drug concentrations for TET and DOX (Appendix Fig S4)."

-In TetB^{L9F}, ~2.5 fold decreases in kappa values are now seen across all drugs. We modified the following text to reflect this.

Previous: "Interestingly, γ remained similar to WT in all drugs while κ changed at most 2-fold in DOX (Appendix Table S1)."

Revised: "Interestingly, γ remained similar to WT, while κ changed 2.5-fold in all drugs (Appendix Table S1)."

-Now variant TetB^{G366C} is shown to change both kappa and gamma in TET and DOX. The following sentence was edited accordingly.

Previous: "This mutation resulted in a change in both κ and γ , however, γ was more severely decreased."

Revised: "This mutation resulted in severe decreases in both κ and γ in all drugs."

In section “The relative change in substrate binding and pumping efficiency rates can be determined after incorporation of protein concentration into the physicochemical-fitness model”

-Now we see comparable changes in relative binding affinity and pumping efficiency of TetB^{L9F} across all drugs. Previously the text read:

Previous: “TetB^{L9F} primarily has an increase in the relative binding affinity in all drugs increasing from TET to DOX to MCN. TetB^{L9F} also shows a slight decrease in pumping efficiency in TET and DOX.”

Revised: “TetB^{L9F} produces both an increase in relative binding affinity and decreases in pumping efficiency across all drugs by comparable magnitudes.”

In section “The physicochemical-fitness model can be used to screen a large plasmid encoded protein library”

Global parameters A and B for host strain *E. coli* BW25113 with empty plasmid pSC101 obtained from growth rates in TET were analyzed after making the background correction and the text has been edited accordingly:

Previous: “These values were calculated to be A=1.65 and B=1.88, and are comparable to the first analysis performed in the absence of empty plasmid (Table 1).”

Revised: These values were calculated to be A=1.91 and B=1.42, and are comparable to the first analysis performed in the absence of empty plasmid (Table 1).”

We no longer highlight mutant V339L in isolation. On average we see similar trends as before with the mutant, but the introduction of the plasmid significantly increases variability in replicates and so changes are harder to distinguish. We now choose to highlight mutant 7 along with mutant 6.

In section “Discussion” we deleted the passage below. As a result of the more accurate background correction suggested by the reviewer, we see a wide spread of effects of the mutation on binding and pumping efficiency across mutants, from no effects on TET, to changes in DOX and MCN.

Previous: “Interestingly, the strain expressing *tet(B)*^{V339C} manifested changes in pumping efficiency, *r*, for each drug but the extent of change was comparable for all drugs suggesting that while there was no change in specificity there were generalized changes in how well this variant could pump across the membrane (Appendix Fig S5). Since the B-helices are postulated to mediate the N- and C- domain interface, it is possible that the mutation V339C is impacting this interface in such a way in TetB that substrate transport is generally altered for TET, DOX, and MCN.”

Revised: “Cells expressing B-helix mutant *tet(B)*^{V339C} showed minimal but statistically significant increases in *k₁* in DOX, no change in TET and could not be resolved for MCN. Interestingly, the strain expressing *tet(B)*^{V339C} manifested changes in pumping efficiency, *r*, for each drug but the extent of change was dissimilar for all drugs suggesting that while there was no change in specificity there were generalized changes in how well this variant could pump across the membrane (Appendix Fig S4). Since the B-helices are postulated to mediate the N- and C- domain interface, it is possible that the mutation V339C is impacting this interface in such a way in TetB that substrate transport is differentially altered for TET, DOX, and MCN.”

Model Curation Report:

Technical curation for the mathematical models in MSB-17-7635R

The equation used for Figure 1 and 2 is well described in the manuscript, and I could readily reproduce the results shown in these figures. It was not clear from the manuscript how the growth rate shown in Figure 3 was calculated but after contacting the authors, and receiving their feedback, I could reproduce the simulation plots in Figure 3.

This is in reference to how the normalization was done. The following text was added to the methods section in the mathematical analysis section:

“To fit the models, the data was then normalized by dividing all data points by the average growth rate at zero drug concentration across the replicates.”

I would advice to indicate in the manuscript how the growth rate as plotted in Figure 3 was calculated, i.e. give an equation or state explicitly how it can be derived from Eq. 1. In addition the authors should refer to the table in which the non inhibited growth rates for the different mutant strains are listed.

We have now included the following text in Material and Methods under Mathematical Analysis to state how the growth rate as a function of drug concentration plotted in Figure 3B was calculated:

“To fit the models, the data was then normalized by dividing all data points by the average growth rate at zero drug concentration across the replicates.”

The source data was originally uploaded to correspond to Appendix Figure S2. The source data is now renamed to correspond with Figure 3.

3rd Editorial Decision

17 November 2017

Thank you again for sending us your revised manuscript. The two reviewers who were asked to evaluate the revised manuscript think that most issues have now been satisfactorily addressed. Reviewer #2 lists a couple of remaining concerns, mostly referring to including further discussion, which we would ask you to address in a minor revision.

REVIEWER REPORTS

Reviewer #1:

The authors have fully addressed the issues that I raised.

Reviewer #2:

The authors have performed additional analysis and in particular improved the determination of the growth rates. It is interesting to see that the latter had a prominent impact on the findings. However, as a minor suggestion, it would be good to show the background corrected growth curves on a log-y plot to corroborate that growth is indeed exponential and report the absolute growth rate (which should be around $\sim 2 \text{ h}^{-1}$ in LB at 37C); since the data are already collected this can be done easily.

In general, I think the manuscript addresses an interesting topic and aims to answer important questions. However, recent studies with a related aim (in particular the study by Deris et al, Science 2013) are currently only briefly mentioned in the beginning. It would be helpful to more clearly emphasize the differences in findings as the Deris paper also connects microscopic parameters to global observables.

The new Western blot is helpful to corroborate the independence of tetB expression from the growth rate. However, it would be good to quantify these measurements (e.g. by extracting the band intensities from the gel images and dividing it by the background corrected absorbance) and report

the numbers. Further, the error of this measurement should be estimated. A potential issue here is that it is usually hard to detect changes in protein level that are smaller than ~2-fold by Western blots while changes of this magnitude may well affect the conclusions from the model. It is certainly possible but in the light of the papers mentioned in my previous reports, somewhat surprising that the expression of tetB from an effectively constitutive promoter appears to be independent of growth rate. I still think that this work would be significantly improved by taking into account the established relations between growth rate and gene expression since this has essentially become standard for this type of modeling. To convince the experts in this field, at the least, it would be helpful to include a discussion why the authors think that these relations do not apply in the present case.

3rd Revision - authors' response

29 November 2017

Point by Point Reviewer Response

Reviewer #1:

The authors have fully addressed the issues that I raised.

Thank you.

Reviewer #2:

The authors have performed additional analysis and in particular improved the determination of the growth rates. It is interesting to see that the latter had a prominent impact on the findings. However, as a minor suggestion, it would be good to show the background corrected growth curves on a log-y plot to corroborate that growth is indeed exponential and report the absolute growth rate (which should be around $\sim 2 \text{ h}^{-1}$ in LB at 37C); since the data are already collected this can be done easily.

Thank you for this suggestion. We have now included Appendix Figure S11 to the manuscript which displays the background corrected logarithmically transformed OD_{600} growth curves for BW25113 expressing $tet(B)^{WT}$ at a range of MCN concentrations (Part A). We also plot the fitness function as the absolute growth rates as a function of selection strength (Part B). We plotted one data set as a visual example since there is too much data to include. Exponential growth can be verified for the rest of the data by plotting the source data included in the paper.

In general, I think the manuscript addresses an interesting topic and aims to answer important questions. However, recent studies with a related aim (in particular the study by Deris et al, Science 2013) are currently only briefly mentioned in the beginning. It would be helpful to more clearly emphasize the differences in findings as the Deris paper also connects microscopic parameters to global observables.

We address this suggestion along with the one below together. Please see our response below.

The new Western blot is helpful to corroborate the independence of tetB expression from the growth rate. However, it would be good to quantify these measurements (e.g. by extracting the band intensities from the gel images and dividing it by the background corrected absorbance) and report the numbers. Further, the error of this measurement should be estimated.

We have included the Western blot again below for visual comparison. Again, the high growth rate relates to a normalized growth rate of ~ 0.6 and the low relates to a normalized growth rate of ~ 0.25 . Using ImageJ, bands were selected, signal area was plotted, and background was subtracted. For *E. coli* BW25113 expressing $tet(B)$ with 20k RBS strength, the high and low growth rate band intensities are 8145 and 8055, respectively. For *E. coli* BW25113 expressing $tet(B)$ with 35k RBS strength, the high and low growth rate band intensities are 24193 and 23249, respectively. These results further show that within our system and experimental set up, there is minimal coupling between growth rate and $tet(B)$ expression.

A potential issue here is that it is usually hard to detect changes in protein level that are smaller than ~2-fold by Western blots while changes of this magnitude may well affect the conclusions from the model. It is certainly possible but in the light of the papers mentioned in my previous reports, somewhat surprising that the expression of tetB from an effectively constitutive promoter appears to be independent of growth rate. I still think that this work would be significantly improved by taking into account the established relations between growth rate and gene expression since this has essentially become standard for this type of modeling. To convince the experts in this field, at the least, it would be helpful to include a discussion why the authors think that these relations do not apply in the present case.

We added sentences regarding our assumption that protein concentrations remain constant despite different growth rates and highlighted the differences between the work in Deris et al. and this paper. For clarity, we changed wording in the second to last paragraph of ‘A physiological model describing TetB efflux pump kinetics can be used to determine physicochemical parameters from fitness’ to highlight that our system is under constant inducible expression instead of constitutive expression.

Previous text: Our host strain BW25113 has a deletion in the genes necessary for arabinose catabolism, therefore, *tet(B)* expression is constitutive in the presence of saturating arabinose and we assume that P_{tot} is constant within our system (Grenier et al, 2014).

Revised/ Added text: Our host strain BW25113 has a deletion in the genes necessary for arabinose catabolism, therefore, *tet(B)* expression is under constant induction in the presence of saturating arabinose and we assume that P_{tot} is constant within our system (Grenier et al, 2014). Previous work has shown that inadvertent coupling between growth rate and gene expression can lead to bistability in growth response of a drug resistant *E. coli* strain in the presence of chloramphenicol (Deris et al, 2013). The authors show this using a phenomenological model, where concentrations of the drug-deactivating enzyme chloramphenicol acetyltransferase are growth rate dependent. Here, we greatly minimize any measurable coupling between expression of TetB and growth rate of the cell through the constant saturating induction of the inducible promoter. This allows us to create a simpler physiological model that can provide more information about the kinetic properties of the protein.

Corresponding Author Name: Professor Yousif Shamoo

Manuscript Number: MSB-17-7635RR